# *How to think step-by-step:* A mechanistic understanding of chain-of-thought reasoning

**Subhabrata Dutta**[*]                                          *subha0009@gmail.com*
*IIT Delhi, India*

**Joykirat Singh**[*]                                          *joykiratsingh18@gmail.com*
*Independent*[†]

**Soumen Chakrabarti**                                          *soumen@cse.iitb.ac.in*
*IIT Bombay, India*

**Tanmoy Chakraborty**[‡]                                          *tanchak@iitd.ac.in*
*IIT Delhi, India*

**Reviewed on OpenReview:** *https://openreview.net/forum?id=uHLDkQVtyC*

## Abstract

Despite superior reasoning prowess demonstrated by Large Language Models (LLMs) with Chain-of-Thought (CoT) prompting, a lack of understanding prevails around the internal mechanisms of the models that facilitate CoT generation. This work investigates the neural sub-structures within LLMs that manifest CoT reasoning from a mechanistic point of view. From an analysis of Llama-2 7B applied to multistep reasoning over fictional ontologies, we demonstrate that LLMs deploy multiple parallel pathways of answer generation for step-by-step reasoning. These parallel pathways provide sequential answers from the input question context as well as the generated CoT. We observe a functional rift in the middle layers of the LLM. Token representations in the initial half remain strongly biased towards the pretraining prior, with the in-context prior taking over in the later half. This internal phase shift manifests in different functional components: attention heads that write the answer token appear in the later half, attention heads that move information along ontological relationships appear in the initial half, and so on. To the best of our knowledge, this is the first attempt towards mechanistic investigation of CoT reasoning in LLMs. The source code and data are made available at `https://github.com/joykirat18/How-To-Think-Step-by-Step`.

## 1 Introduction

Recent advancements with Large Language Models (LLMs) demonstrate their remarkable reasoning prowess with natural language across a diverse set of problems (OpenAI, 2024; Kojima et al., 2022; Chen et al., 2023). Yet, existing literature fails to describe the neural mechanism within the model that implements those abilities – how they emerge from the training dynamics and why they are often brittle against even unrelated changes. One of these capabilities of LLMs that has boosted their potential in complex reasoning is *Chain-of-Thought* (CoT) prompting  (Wei et al., 2022b; Kojima et al., 2022). Instead of providing a direct answer to the question, in CoT prompting, we expect the model to generate a verbose response, adopting a step-by-step reasoning process to reach the answer. Despite the success of eliciting intermediate computation and their diverse, structured demonstration, the exact mechanism of CoT prompting remains mysterious. Prior attempts have been made to restrict the problem within structured, synthetic reasoning to observe

---
[*]Equal contribution.
[†]Work done as Research Assistant at IIT Bombay
[‡]Corresponding Author

the LLM's CoT generation behavior (Saparov & He, 2023); context perturbation towards causal modeling has also been used (Tan, 2023). A few recent approaches seek to associate the emergence of CoT reasoning ability in LLMs with localized structures in the pertaining data (Prystawski et al., 2024; Wang & Wang, 2023). However, these endeavors produce only an indirect observation of the mechanism; the underlying neural algorithm implemented by the LLM remains in the dark.

Recent developments in the mechanistic interpretation of Transformer-based models provide hope for uncovering the neural 'algorithms' at work inside LLMs (Elhage et al., 2021; Nanda, 2022). Typically, mechanistic methods seek to build a causal description of the model; starting from the output, they localize important components of the model (e.g., an attention head or a multilayer perceptron (MLP) block) by activation patching (Wang et al., 2023; Zhang & Nanda, 2023). However, there are some implicit challenges in reverse-engineering CoT-prompting in foundational models. The capability of generating CoT is largely dependent on model scale (Wei et al., 2022a; Saparov & He, 2023). On the other hand, reverse-engineering a large model becomes a wild goose chase due to the *hydra effect* (McGrath et al., 2023) — neural algorithmic components within LLMs are adaptive: once we 'switch off' one functional component, others will pitch in to supply the missing functionality. Furthermore, in most real-world problems involving multi-step reasoning, there are implicit knowledge requirements. LLMs tend to memorize factual associations from pretraining as key-value caches using the MLP blocks Geva et al. (2021); Meng et al. (2022). Due to the large number of parameters in MLP blocks and their implicit polysemanticity, interpretation becomes extremely challenging.

In this paper, we seek to address these challenges, thus shedding light on the internal mechanism of Transformer-based LLMs while they perform CoT-based reasoning. Going beyond the typical toy model regime of mechanistic interpretation, we work with Llama-2 7B (Touvron et al., 2023). To minimize the effects of MLP blocks and focus primarily on reasoning from the provided context, we make use of the PrOntoQA dataset (Saparov & He, 2023) that employs ontology-based question answering using fictional entities (see Figure 1 for an example). Specifically, we dissect CoT-based reasoning on fictional reasoning as a composition of a fixed number of subtasks that require decision-making, copying, and inductive reasoning (☛ Section 4).

We draw on three prominent techniques for investigating neural algorithms, namely, activation patching (Nanda, 2022), probing classifiers (Belinkov, 2022), and logit lens (nostalgebraist, 2020) to disentangle different aspects of the neural algorithm implemented by Llama-2 7B. We find that despite the difference in reasoning requirement of different subtasks, the sets of attention heads that implement their respective algorithms enjoy significant intersection (☛ Section 4.1). Moreover, they point towards the existence of mechanisms similar to induction heads (Olsson et al., 2022) working together. In the initial layers of the model (typically, from first to 16th decoder blocks in Llama-2 7B), attention heads conduct information transfer between ontologically related fictional entities; e.g., for the input *numpuses are rompuses*, this mechanism copies information from *numpus* to *rompus* so that any pattern involving the former can be induced with the latter (☛ Section 5). Interestingly, in this same segment of the model, we find a gradual transition in the residual stream representation from pretraining before in-context prior, i.e., the contextual information replaces the bigram associations gathered via pretraining (☛ Section 6.1).

Following this, we seek to identify the pathways of information that are responsible for processing the answer and write it to the output residual stream for each subtask. To deal with the abundance of backup circuits (Wang et al., 2023), we look for such parallel pathways simultaneously without switching off any of them. We find that multiple attention heads simultaneously write the answer token into the output, though all of them appear at or after the 16th decoder block (☛ Section 6.2). These answer tokens are collected from multiple sources as well (i.e., from the few-shot context, input question, and the generated CoT context), pointing towards the coexistence of multiple neural algorithms working in parallel.

Our findings supply empirical answers to a pertinent open question about whether LLMs actually rely on CoT to answer questions (Tan, 2023; Lampinen et al., 2022): the usage of generated CoT varies across subtasks, and there exists parallel pathways of answer collection from CoT as well as directly from the question context (☛ Section 6.3). Here again, we observe the peculiar functional rift within the middle of the model: answer tokens, that are present in the few-shot examples but contextually different from the same token in the question, are used as sources by attention heads, primarily before the 16th decoder block.

To the best of our knowledge, this is the first-ever in-depth analysis of CoT-mediated reasoning in LLMs in terms of the neural functional components.

## 2    Related work

In this section, we provide an overview of literature related to this work, primarily around different forms of CoT reasoning with Transformer-based LMs, theoretical and empirical investigations into characteristics of CoT, and mechanistic interpretability literature around language models.

Nye et al. (2021) first showed that, instead of asking to answer directly, letting an autoregressive Transformer generate intermediate computation, which they called *scratchpads*, elicits superior reasoning performance. They intuitively explained such behavior based on the observation that, in a constant depth-width model with $\mathcal{O}(n^2)$ complexity attention, it is not possible to emulate an algorithm that requires super-polynomial computation; by writing the intermediate answer, this bottleneck is bypassed. Wei et al. (2022b) demonstrated that LLMs enabled with unstructured natural language expressions of intermediate computations, aka CoT, can demonstrate versatile reasoning capabilities.

Multiple recent attempts have been made toward a deeper understanding of CoT, both empirically and theoretically. Feng et al. (2023) employed circuit complexity theory to prove that — (i) Transformers cannot solve arithmetic problems with direct answers unless the model depth grows super-polynomially with respect to problem input size, and (ii) Constant depth Transformers with CoT generation are able to overcome the said challenge. Liu et al. (2022) showed that Transformers can learn *shortcut* solutions to simulate automata with standard training (without CoT); however, they are statistically brittle, and recency-biased scratchpad training similar to that of Nye et al. (2021) can help with better generalizability and robustness. Saparov & He (2023) dissected the behavior of LLMs across scale on synthetically generated multistep reasoning tasks on true, false, and fictional ontologies. In a somewhat counterintuitive finding, Lampinen et al. (2022) observed that even if an LLM is prompted to generate an answer followed by an explanation, there is a significant improvement over generating an answer without any explanation. While this might point toward a non-causal dependence between the explanation and the answer, Tan (2023) showed that LLMs do utilize, at least partially, the information generated within the intermediate steps such that the answer is causally dependent on the CoT steps. Prystawski et al. (2024) theoretically associated the emergence of CoT with localized statistical structures within the pertaining data. Wang & Wang (2023) investigated reasoning over knowledge graph to characterize the emergence of CoT. They identified that reasoning capabilities are primarily acquired in the pertaining stage and not downstream fine-tuning while empirically validating Prystawski et al. (2024)'s theoretical framework in a more grounded manner. Bi et al. (2024) found that while reasoning via program generation, LLMs tend to favor an optimal complexity of the programs for best performance.

The above-mentioned studies either 1) dissect the model behavior under controllable perturbations in the input problem or 2) construct theoretical frameworks under different assumptions to prove the existence of certain abilities/disabilities. Mechanistic interpretability techniques go one step deeper and seek to uncover the neural algorithm deployed by the model to perform a certain task. Elhage et al. (2021) studied one-, two-, and three-layer deep, attention-only autoregressive Transformers and their training dynamics. A crucial finding of their investigation was the existence of *induction heads* — a composition of two attention heads at different layers that can perform pattern copying from context. Olsson et al. (2022) empirically observed the simultaneous emergence of induction heads and in-context learning in the training dynamics of similar toy Transformer models. Similar analyses on phenomena like polysemanticity, superposition, and memorization have been performed in the toy model regime (Elhage et al., 2022; Henighan et al., 2023). Beyond these toy models, Wang et al. (2023) analyzed circuits in GPT-2 responsible for Indirect Object Identification. Wu et al. (2023) proposed a causal abstraction model to explain price-tagging in the Alpaca-7B model: given a range of prices and a candidate price, the task is to classify if the candidate price falls within the given range. Extrapolating mechanistic observations from the regime of toy models to 'production' LLMs is particularly challenging; with billions of parameters and deep stacking of attention heads, identifying head compositions that instantiate an algorithm is extremely difficult. Furthermore, as McGrath et al. (2023) suggest, different components of the LLM are loosely coupled and adaptive in nature, for which they coined the term *hydra*

*effect* — ablating an attention layer may get functionally compensated by another. Wang et al. (2023) also identified similar information pathways called *backup circuits* that can take over once the primary circuits are corrupted.

## 3 Background

In this section, for completeness, we briefly introduce the concepts and assumptions necessary for delving into the interpretation of CoT reasoning.

The **Transformer architecture** (Vaswani et al., 2017) consists of multiple, alternated attention and MLP blocks, preceded and followed by an embedding projection and a logit projection, respectively. Given the vocabulary as $V$, we denote the embedded representation of the $i$-th token $s_i \in V$ in an input token sequence $S$ of length $N$ (i.e., the sum of embedding projections and position embeddings in case of additive position encoding) as $\boldsymbol{x}_0^i \in \mathbb{R}^d$, where $d$ is the model dimension. We denote the content of the residual stream corresponding to the $i$-th token, input to the $j$-th decoder block, as $\boldsymbol{x}_{j-1}^i$, which becomes $\tilde{\boldsymbol{x}}_{j-1}^i$ after the attention layer reads and writes on the residual stream. The initial content of the residual stream is the same as the token embedding $\boldsymbol{x}_0^i$. Assuming $H$ number of heads for each attention layer, the operation of $k$-th attention head at $j$-th decoder block on the $i$-th token's residual stream is denoted as $\boldsymbol{y}_{j,k}^i = h_{j,k}(\boldsymbol{x}_{j-1}^i)$, $\boldsymbol{y}_{j,k}^i \in \mathbb{R}^d$. Then $\tilde{\boldsymbol{x}}_j^i = \boldsymbol{x}_{j-1}^i + \sum_k \boldsymbol{y}_{j,k}^i$ is the content of the residual stream immediately after the attention operations. Each attention head $h_{j,k}$ is parameterized using four projection matrices: query, key and value projection matrices, denoted by $\boldsymbol{W}_Q^{j,k}, \boldsymbol{W}_K^{j,k}, \boldsymbol{W}_V^{j,k} \in \mathbb{R}^{d \times \frac{d}{H}}$, respectively, and output projection matrix $\boldsymbol{W}_O^{j,k} \in \mathbb{R}^{\frac{d}{H} \times d}$. Note that in the case of Llama models, the position encoding is incorporated via rotation of query and key projections before computing dot-product (Su et al., 2023); we omit this step for brevity. Similarly, the action of the MLP block can be expressed as $\boldsymbol{z}_j^i = \text{MLP}_j(\tilde{\boldsymbol{x}}_j^i)$, with $\boldsymbol{x}_j^i = \tilde{\boldsymbol{x}}_j^i + \boldsymbol{z}_j^i$ denoting the content of the residual stream after decoder block $j$. After processing the information through $L$ number of decoder blocks, a feedforward transformation $\boldsymbol{U} \in \mathbb{R}^{d \times |V|}$, commonly known as the *unembedding* projection, maps the content of the residual stream into the logit space (i.e., a distribution over the token space). Given a sequence of input tokens $S = \{s_1, \cdots, s_i\}$, the autoregressive Transformer model predicts a new output token $s_{i+1}$:

$$s_{i+1} = \underset{\boldsymbol{x}_{\text{logit}}^i}{\arg\max} \, \text{LM}(\boldsymbol{x}_{\text{logit}}^i | S) \tag{1}$$

**Fictional ontology**-based reasoning, as proposed by Saparov & He (2023) as PrOntoQA, provides a tractable approach to dissect CoT generation. The reasoning problem is framed as question-answering on a tree-based ontology of fictional entities (see Appendix A for examples of ontologies and reasoning problems framed). This eases two major challenges in our case: (i) Mechanistic interpretation requires input or activation perturbation and recording the results of such perturbations. With CoT, one then needs to repeat such perturbation process for all the constituent subtasks. Unlike free-form CoT reasoning, PrOntoQA provides a clearly demarcated sequence of successive steps that can be analyzed independently. (ii) The solution to most real-world reasoning problems heavily requires factual knowledge, so much so that a sound reasoning process might get misled by incorrect fact retrieval. A fictional ontology ensures zero interference between the entity relationships presented in the question and the world-knowledge acquired by the model in the pertaining stage. This further minimizes the involvement of the MLP layers in the neural mechanisms implemented by the model as they typically serves as the parametric fact memory of an LM (Geva et al., 2021; Meng et al., 2022). Additionally, PrOntoQA provides reasoning formulation over false ontologies (see example in Appendix A). False ontology grounds the reasoning over statements that are false in the real world. An important demarcation between fictional and false ontological reasoning is that while the former minimizes the effects of factual associations memorized as pretraining prior, the latter requires the LM to actively eclipse such memorized knowledge to solve the problem successfully.

**Circuits**, in mechanistic interpretability research, provide the abstractions of interpretable algorithms implemented by the model within itself. Typically, a *circuit* is a subgraph of the complete computational graph of the model, responsible for a specific set of tasks. We primarily follow the notation adopted by Wang et al.

(2023), with nodes defined by model components like attention heads and projections and edges defined by interactions between such components in terms of attention, residual streams, etc.

**Activation patching** is a common method in interpretability research. Activation patching begins with two forward passes of the model, one with the actual input and another with a selectively corrupted one. The choice of input corruption depends on the task and the type of interpretation required. For example, consider the following Indirect Object Identification (IOI) task (Wang et al., 2023): given an input *John and Mary went to the park. John passed the bottle to*, the model should predict *Mary*. Further, corrupting the input by replacing *Mary* with *Anne* would result in the output changing to *Anne*. Let $\boldsymbol{x}_j^{\text{Mary}}$ and $\boldsymbol{x}_j^{\text{Anne}}$ represent the original and corrupted residual streams at decoder block $j$, depending on whether *Mary* or *Anne* was injected at the input. Now, in a corrupted forward pass, for a given $j$, if the replacement of $\boldsymbol{x}_j^{\text{Anne}}$ by $\boldsymbol{x}_j^{\text{Mary}}$ results in the restoration of the output token from *Anne* to *Mary*, then one can conclude that attention mechanism at decoder block $j$ is responsible for moving the name information. This is an example of patching *corrupted-to-clean* activation, often called *causal tracing* (Meng et al., 2022). Activation patching, in general, refers to both clean-to-corrupted as well as corrupted-to-clean patching (Zhang & Nanda, 2023).

**Knockout** is a method to prune nodes in the full computational graph of the model to identify task-specific circuits. Complete ablation of a node is equivalent to replacing the node output with an all-zero vector. Our initial experiments suggest that such an ablation destructively interferes with the model's computation. Instead, we follow Wang et al. (2023) for *mean-ablation* to perform knockouts. Specifically, we construct inputs from the false ontologies provided in the PrOntoQA dataset and compute the mean activations for each layer across different inputs:

$$\boldsymbol{x}_j^{\text{Knock}} = \text{Mean}_i \left( \{ \boldsymbol{x}_j^i | s_i \in S \in \mathcal{D}_{\text{False}} \} \right) \tag{2}$$

where $\mathcal{D}_{\text{False}}$ denotes the false ontology dataset. Then, the language model function with head $h_{j,k}$ knocked out for the residual stream corresponding to the $l$-th token, can be represented as,

$$s_i^{\text{Knock}} = \arg\max_{\boldsymbol{x}_{\text{logit}}^i} \text{LM}_{j,k}^l (\boldsymbol{x}_{\text{logit}}^i | S, \boldsymbol{y}_{j,k}^l = x_j^{\text{Knock}}) \tag{3}$$

More often than not in this work, we will need to knock out a set of heads $\mathcal{H}$; we will denote the corresponding language model as $\text{LM}_{\mathcal{H}}^l$. Also, if we perform knockout on the last residual stream (which, when projected to the token space, gives us the output token at the current step of generation), we will drop the superscript $l$.

## 4 Task composition

We seek to discover the circuits responsible in Llama-2 7B for few-shot CoT reasoning on the examples from PrOntoQA fictional ontology problems. Consider the example presented in Figure 1: we ask the model "*Tumpuses are bright. Lempuses are tumpuses. Max is a lempus. True or False: Max is bright*" (for brevity, we omit the few-shot examples that precede and the CoT prompt *Let's think step by step* that follows the question). Llama-2 7B generates a verbose CoT reasoning sequence as follows: "*Max is a lempus. Lempuses are tumpuses. Max is a tumpus. Tumpuses are bright. Max is bright.*" We can observe that such a reasoning process constitutes three critical kinds of subtasks:

1. **Decision-making**: The model decides on the path of reasoning to follow. In Figure 1, given the three possible entities to start with, namely *Tumpus*, *Lempus*, and *Max*, Llama-2 7B starts with the last one (i.e., *Max is a lempus*). One would require multiple such decision-making steps within the complete reasoning process.

2. **Copying**: The LM needs to copy key information given in the input to the output. Typically, decision-making precedes copying, as the model needs to decide which information to copy.

3. **Induction**: The LM uses a set of statements to infer new relations. Again, a decision-making step precedes as the model must decide on the relation to infer. In the majority of this work, we focus on 2-hop inductions where given statements of the form `A is B` and `B is C`, the model should infer `A is C`.

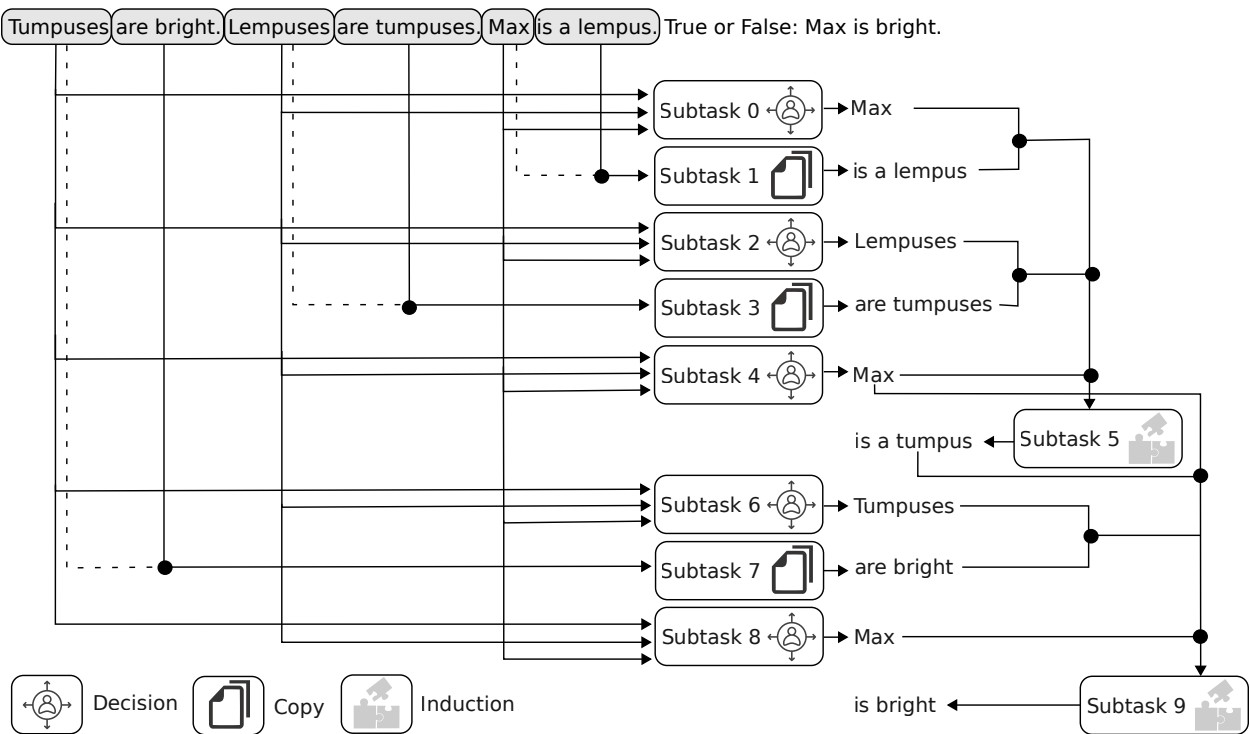

Figure 1: **An working example of task decomposition of CoT generation on fictional ontology**. *Decision*-type subtasks (0, 2, 4, 6, and 8) choose which reasoning path to follow; such decisions may be about the order of the information to be copied from the question to the generated output or whether inductive reasoning will be performed. *Copy*-type subtasks (1, 3, and 7) follow Decision subtasks and copy statements from the context to output. *Induction*-type subtasks (5 and 9) perform the reasoning steps of the form "if A is B and B is C, then A is C".

In all our analyses henceforth, we follow a structure where the model needs to generate CoT responses that solve the overall task using ten steps or subtasks, each belonging to one of the three categories, as presented in Figure 1. (See Appendix E for details of few-shot examples used along with the overall performance of Llama-2 7B.) A natural hypothesis would be that distinct, well-defined circuits corresponding to each category of subtasks (i.e., decision-making, copying, and inductive reasoning) exist that work in conjunction to lead the CoT generation. However, as our findings in the next section suggest, this is not the case in reality.

## 4.1 Task-specific head identification

As Wang et al. (2023) suggested, the very first step toward circuit discovery is to identify the components. Since we presume the attention heads as the nodes of a circuit, the goal then becomes to identify those heads in the language model that are most important for a given task. We define the importance of the $k$-th head at the $j$-th decoder block, $h_{j,k}$ for a particular task as follows.

Let the task be defined as predicting a token $s_{i+1}$ given the input $S = \{s_1, s_2, \cdots, s_i\}$. For example, in the demonstration provided in Figure 1, predicting the token *Max* given the context *Tumpuses are bright. Lempuses are tumpuses. Max is a lempus. True or False: Max is bright. Response: Let us think step by step. Tumpuses are bright. Lempuses are tumpuses.* We assign a score $\mu_{\text{Task}}(h_{j,k})$ to each head $h_{j,k}$ proportional to their importance in performing the given task. Following the intuitive subtask demarcation presented earlier, we start with scoring the attention heads for each different subtask. We provide the detailed procedure of calculating $\mu_{\text{Task}}(h_{j,k})$ for each category of subtasks — decision-making, copying, and induction, in Appendix C.

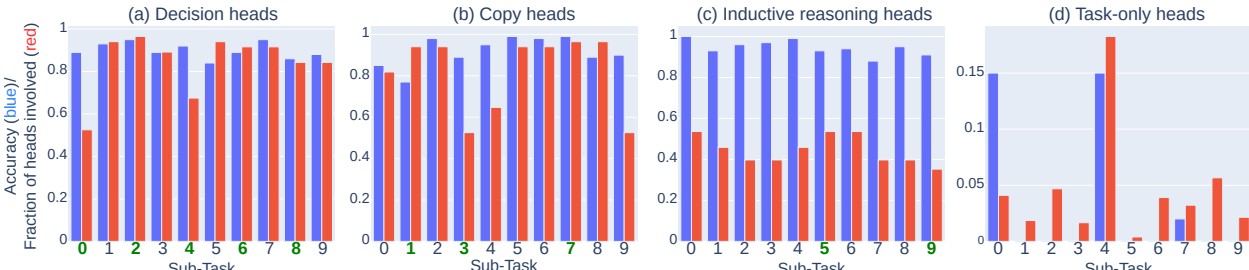

Figure 2: **Performance of attention heads identified for each different subtask type across different subtasks.** We show the performance of (a) decision-making, (b) copy, and (c) inductive reasoning heads for each subtask 0 to 9 (blue bars show accuracy when the rest of the heads are knocked out; red bars denote the fraction of heads involved). Bold green indices denote the subtasks belonging to the task category used for attention head removal. (d) Task-only heads are only those that are not shared with other tasks, e.g., only those copying heads for subtask 4 that are not decision-making or inductive reasoning heads. Inductive reasoning heads are consistently functional across all the subtasks with the least number of heads involved.

Figure 2 demonstrates how each of these three categories of attention heads performs on its respective subtasks and the rest of the subtasks (note that we perform head ablation here for each subtask index independently). We also show the fraction of the total number of heads involved in performing each subtask. A higher accuracy with a lower number of head involvement would suggest that we have found the minimal set of heads constitute the responsible circuit. As we can observe, there are no obvious mappings between the tasks used for head identification and the subtasks that constitute the CoT. Quite surprisingly, heads that we deem responsible for inductive reasoning can perform well across all the subtasks (Figure 2 (c)) and requires the least number of active heads across all tasks. Furthermore, the three sets of heads share a significant number of heads that are essential for the subtasks. In Figure 2(d), for each subtask, we use the heads that are not shared by subtasks of other categories. For example, in subtask 2, since we assumed it to be a decision-making subtask, we took only those decision-making heads that were not present in the copying or inductive reasoning heads. It is evident that these tasks are not structurally well differentiated in the language model. A good majority of heads share the importance of all three subtasks. The existence of backup circuits justifies such phenomena; however, with large models, the "self-repairing" tendency (McGrath et al., 2023) is much higher. We hypothesize that there exists even further granularity of functional components that collectively express all three types of surface-level subtasks. We seek a solution in the earlier claims that pattern matching via induction heads serves as the progenitor of in-context learning (Olsson et al., 2022). We observe that the inductive reasoning subtasks of the form if `[A]` is `[B]` and `[B]` is `[C]` then `[A]` is `[C]` essentially requires circuits that can perform 1) representation mixing from `[A]` to `[B]` to `[C]`, and 2) pattern matching over past context by *deciding* which token to copy and then copying them to the output residual stream. Therefore, heads responsible for inductive reasoning can perform decision-making and copying by this argument.

For empirical validation of this argument, we analyze the effects of knocking out heads on the accuracy of each of the subtask indices. For each subtask, we construct a histogram of head counts over $\mu(h_{j,k})$. It can be seen that the corresponding density distribution is Gaussian in nature. Furthermore, we observe that the actual effect of pruning a head on the accuracy is not dependent on the actual value of $\mu(h_{j,k})$ but varies with $\delta(h_{j,k}) = (\mu(h_{j,k}) - \text{Mean}_{j,k}(\mu(h_{j,k}))^2$. From the definition of $\mu(h_{j,k})$ provided in Appendix C, we can explain the behavior as follows: a very low value of $\mu(h_{j,k})$ is possible when the KL-divergence between the noisy logits and original logits is almost same as the KL-divergence between patched and original logits. Conversely, $\mu(h_{j,k})$ can be very high when the KL-divergence between the patched logit and the original logit is very small. Both of these suggest a strong contribution of the particular head.

Figure 3 shows the accuracy of the model at each different subtask indices with heads pruned at different $\delta(h_{j,k})$ (we show the histogram over $\mu(h_{j,k})$ and start pruning heads from the central bin corresponding to the smallest values of $\delta(h_{j,k})$). Among the 1,024 heads of Llama-2 7B, we can see that pruning the model

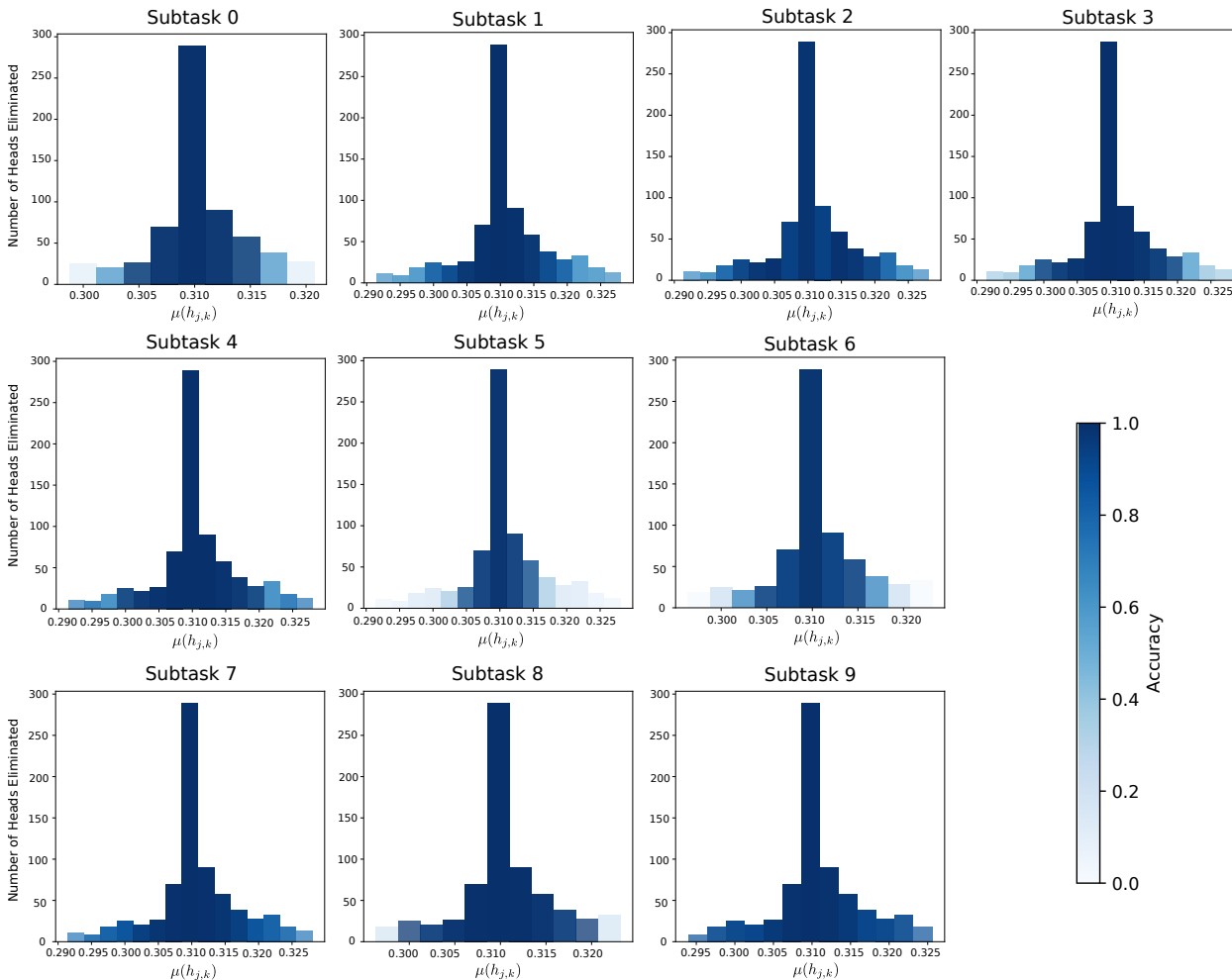

Figure 3: **Importance of inductive reasoning heads across subtasks**. We plot the counts of attention heads binned over relative head importance ($\mu(h_{j,k})$) computed on inductive reasoning task across all subtask indices (0-9, left to right and top to bottom). For a given subtask and a given bin of attention heads (corresponding to a range of $\mu(h_{j,k})$) knocked out, we show the accuracy of the pruned model relative to the original model (color-coded in blue). A wider spread of dark-colored bins for a subtask signifies that a higher number of attention heads can be knocked out without hurting the model performance; for example, in subtask 7, the removal of attention heads with $\mu(h_{j,k}) \in [0.295, 0.325]$ retains a similar performance compared to heads removed in subtask 5 with $\mu(h_{j,k}) \in [0.305, 0.315]$, signifying that more attention heads are required to remain active in subtask 5 compared to 7.

to as low as $\sim$400 heads retains $>90\%$ of the original accuracy (see Table 1 in Appendix for subtask-wise statistics).

Further patterns in allocating attention heads for different subtasks can be observed in Figure 3. Typically, subtask indices 0 and 5 are most sensitive to pruning attention heads; the accuracy of the model quickly decreases as we knock out attention heads with $\mu(h_{j,k})$ deviated from the mean value. Subtask-0 is a decision-making subtask where the model decides on the reasoning path. One can further demarcate between subtasks 0 vs 2, 6, and 8, while we initially assumed all of these four to be decision-making stages. Referring to the example provided in Figure 1, in subtask-0, the model has very little information about which ontological statement to choose next, compared to subtask indices 2, 6, and 8, where the model needs to search for the fact related to the second entity in the last generated fact. Subtask 5 is the actual inductive reasoning step.

Compared to subtask 9, which is also an inductive reasoning step, subtask 5 does not have any prior (the answer generated in subtask 9 is framed as a question in the input).

# 5 Token mixing over fictional ontology

In the earlier section, we observed that the set of heads that we identify as responsible for inductive reasoning performs consistently across all subtasks with the least number of heads remaining active, retaining more than 90% of the original performance with as low as 40% of the total number of heads. We continue to investigate the inductive reasoning functionality with this reduced set of heads. Recall that we first need to explore if there is indeed a mixing of information happening across tokens of the form `[A] is [B]`. Specifically, we pose the following question – given the three pairs of tokens ($[A_1]$, $[B_1]$), ($[A_2]$, $[B_2]$) and ($[A_3]$, $[B_3]$), such that there exists a positive relation `[A₁] is [B₁]`, a negative relation `[A₂] is not [B₂]`, and no relations exist between `[A₃]` and `[B₃]`, is it possible to distinguish between $[\boldsymbol{x}_j^{A_1} : \boldsymbol{x}_j^{B_1}]$, $[\boldsymbol{x}_j^{A_2} : \boldsymbol{x}_j^{B_2}]$, and $[\boldsymbol{x}_j^{A_3} : \boldsymbol{x}_j^{B_3}]$, where $[\cdot : \cdot]$ denotes the concatenation operator, for a given decoder layer $j$? Given that these tokens denote entities from a fictional ontology, the model could not possibly have memorized similar representations with the embedding/MLP blocks for the tokens. Instead, it needs to deploy the attention heads to move information from the residual stream of one token.

We translate the posed question into learning a probing classifier (Belinkov, 2022) $\mathcal{C} : \mathcal{X} \to \mathcal{Y}$, where for each $[\boldsymbol{x}_j^{A_i} : \boldsymbol{x}_j^{B_i}] \in \mathcal{X}$, the corresponding label $y \in \mathcal{Y} = \{-1, 0, +1\}^{|\mathcal{X}|}$ should be $-1$, $0$, or $+1$, if $A_i$ and $B_i$ are negatively related, unrelated, or positively related, respectively. There can be multiple occurrences of a token in the input context (e.g., there are two occurrences of [B] in the above example). As a result, there will be multiple residual streams emanating from the same token. To decide which residual streams to pair with (in positive/negative pairing), we rely on immediate occurrence. For example, given the input `[A] is [B]. [A] is not [C]`, we take the first occurrence of [A] to be positively related to [B], while the second occurrence of [A] as negatively related to [C]. We experiment with a linear and a non-linear implementation of $\mathcal{C}$ as feedforward networks. Specifically, in the linear setting, we use a single linear transformation of the form $y = (\boldsymbol{W}\boldsymbol{x} + \boldsymbol{B})$ followed by a softmax; in the nonlinear setup, we utilize stacked linear transformations with ReLU as intermediate non-linearity. We provide the full implementation details of this experiment in Appendix D. Given that the baseline distribution of the residual stream representation might vary across layers, we employ layer-specific classifiers.

Figure 4 shows the 3-way classification performance in terms of accuracy across different layers using 4-layer ReLU networks. The following conclusions can be drawn from the observations:

- **Non-linear information movement between residual streams.** Our experiments with the linear classifier fail to distinguish between ontologically related, unrelated, and negatively related entities.

- **Distinguishability of the residual stream pairs improves across depth.** While the very first layer of attention provides a substantial degree of required token mixing, we observe that the classification performance goes better gradually, pointing towards the existence of multiple successive attention heads that continue moving information from $A_j$ to $B_j$ if they are related. However, after a certain depth, the distinguishability starts diminishing, most likely due to accumulation of other information related to the tasks.

- **Token mixing is not boosted from contextual prior.** Across different numbers of in-context examples provided, we do not observe any significant difference in peak classification performance compared to zero-shot regimes. However, as the number of examples increases, we observe unstable classification performance. It is likely that with more in-context examples, task-specific information is accumulated earlier than the zero-shot regime.

# 6 Circuitry of step-by-step generation

In Section 4.1, we observed that the head importance identified via inductive reasoning task could serve as a proxy to identify the important heads across all the subtasks. This provides us with an opportunity to reduce

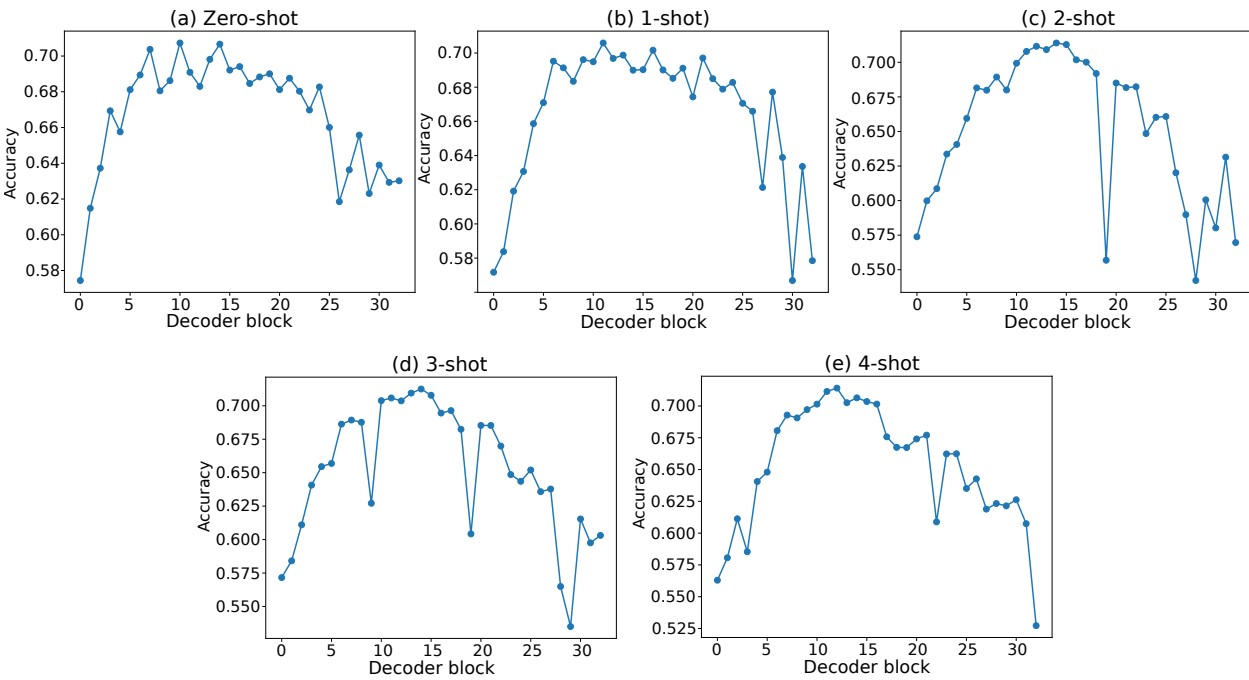

Figure 4: **How does the LLM mix information among tokens according to ontology?** We plot the performance (in terms of accuracy) of classifying whether two tokens are ontologically related, unrelated, or negatively related using their residual stream representations at different layers. We demonstrate the depth-wise accuracy profile using different numbers of few-shot examples (0 to 4). Typically, mixing information between tokens according to their relation does not require in-context learning. Information mixing between related (or negatively related) token pairs results in better distinguishability: gradually increasing from the starting decoder blocks and achieving a peak between decoder blocks 10-15.

the total number of attention heads to analyze when investigating the step-by-step generation procedure. For each subtask, we use a threshold range for $\mu_{h,k}$ to select a subset of attention heads that are kept intact while the rest of the heads are knocked out (see Appendix C for the subtask-wise threshold range used and the corresponding subtask accuracy). This chosen subset of the model gives us an aggregate accuracy of 0.9 over the inputs for which the full model generates correctly.

Next, we proceed to look into the exact information that is being read and written by the attention heads from and into the residual streams. Given a head $h_{j,k}$ that writes $\boldsymbol{y}_{j,k}^i$ to the residual stream $\boldsymbol{x}_{j-1}^i$, we apply the unembedding projection $\boldsymbol{U}$ on $\boldsymbol{y}_{j,k}^i$ and select the token with the highest probability, $\hat{s}_i^{j,k}$. The unembedding projection provides the opportunity to look into the token-space representation directly associated with the residual stream and its subspaces (nostalgebraist, 2020). However, it should be noted that applying unembedding projection typically corresponds to Bigram modeling (Elhage et al., 2021). Therefore, when we map any intermediate representation (attention output, residual stream, etc.), we essentially retrieve the token that is most likely to follow.

## 6.1 In-context prior vs pretraining prior

We start by exploring the depth at which the model starts following the context provided as input. Specifically, we check for a given token $s_i$ in the sequence $S$, if $\hat{s}_i^{j,k}$, the token projected by the attention head $h_{j,k}$ is such that $\langle s_i, \hat{s}_i^{j,k} \rangle$ is bigram present in $S$. We compute a *context-abidance score*, $c_{j,k}$ for the head $h_{j,k}$ as the fraction of tokens for which the above condition holds true.

In Figure 5, we plot the context abidance distribution for each head across different subtask indices. We can observe a visible correlation between the depth of the head and how much context abiding it is —

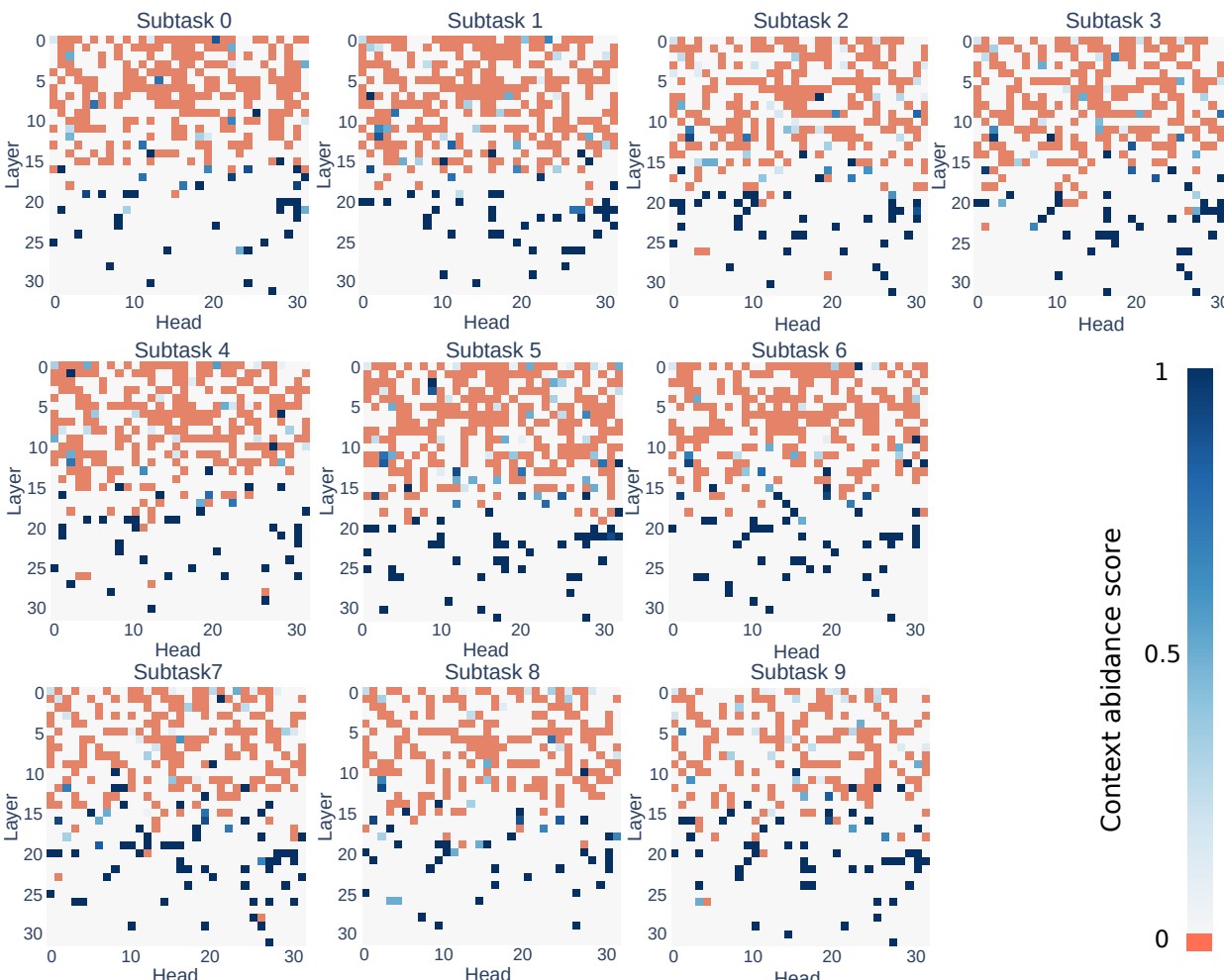

Figure 5: **When does the LLM start following the context?** Distribution of context-abidance score $c_{j,k}$ of each head (darker blue shade signifies higher $c_{j,k}$) with layer ($j$) on y-axis and head index ($k$) on x-axis. Attention heads in red denote those with zero context-abidance. We show the distributions for all the subtask indices 0-9 (left to right, top to bottom). Typically, context abidance is task-agnostic and emerges after the 16th decoder block.

*the LM starts focusing on the contextual information at deeper layers.* Given that our experiment setup is based on fictional ontology, $c_{j,k}$ provides a strong demarcation between pretraining prior (i.e., language model statistics memorized from pretraining) and contextual prior (i.e., language model statistics inferred from context) since there are negligible chances of the LM to memorize any language modeling statistics containing the fictional entities from the pretraining data. Therefore, it predicts a correct bigram only when it is able to follow the context properly. However, there is no visible demarcation across different subtasks; we can claim that this *depth-dependence is implicit to the model and does not depend on the prediction task.*

## 6.2 Subtask-wise answer generation

We investigate the information propagation pathway through the attention heads that constitute the step-by-step answering of the subtasks. We start with the heads that directly write the answer to the particular subtask into the last residual stream by mapping their output to the token space using $U$. Figure 6 plots these answer-writing heads along with the probability of the answer in the attention head output across different subtasks. The existence of multiple answer-writing heads suggests that LMs exploit multiple pathways to generate the same answer, with each such pathway reinforcing the generated output. Looking at the heads

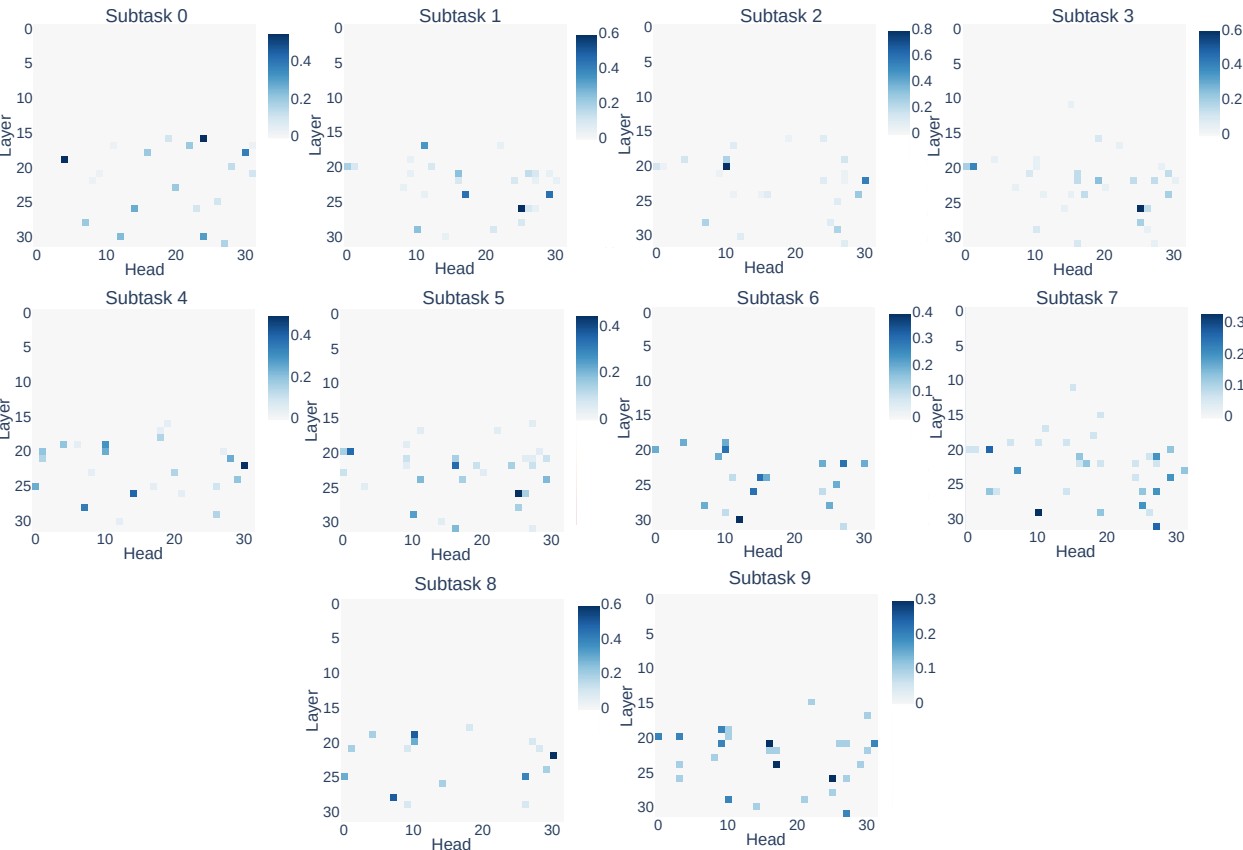

Figure 6: **Which heads are writing the answer?** For each subtask (0-9), we show the attention heads that write the answer to the corresponding task into the last residual stream, along with the probability of the answer token in the attention head output.

with the highest answer probability, we can observe that some top heads are utilized across different subtasks, e.g., subtasks 1 and 3, subtasks 2, 4, and 8, subtasks 5 and 9, etc.

**Sharp change in depth-wise functionality.** Interestingly, the 16-th layer appears as a *region of functional transition* in the model with all the answer-writing heads appearing *after* this particular layer. Our earlier experiments also associate this layer with some phase shifts; in the token-mixing analysis (Section 5), the distinguishability of ontologically related entities is observed to achieve peaks near this layer and starts falling after that; in Section 6.1 as well, we see a certain overall rise in context-abidance in the heads after the 16-th layer (see Figure 5). All three findings suggest a *functional rift* in LLMs that happens to lie almost at the halfway point from embedding to unembedding. The embedding layer associates tokens with information available from pretraining. The initial half assists information movement between residual streams and *aligns* the representations to the contextual prior. The latter half of the model employs multiple pathways to write the answer to the last residual stream.

### 6.3  Parallel pathways of answer processing

Once we observe the existence of multiple answer-writers within the model, the natural step forward is to wonder if they all process the answer from the input using the same mechanism. We employ a recursive strategy to identify the flow of information through the attention heads (see Appendix G for a detailed description of the procedure). Specifically, we start from the answer-writing heads, follow which residual streams are being attended by these heads, identify the content of these residual streams via unembedding projection, and identify the heads in the previous layers that are writing that content into those residual

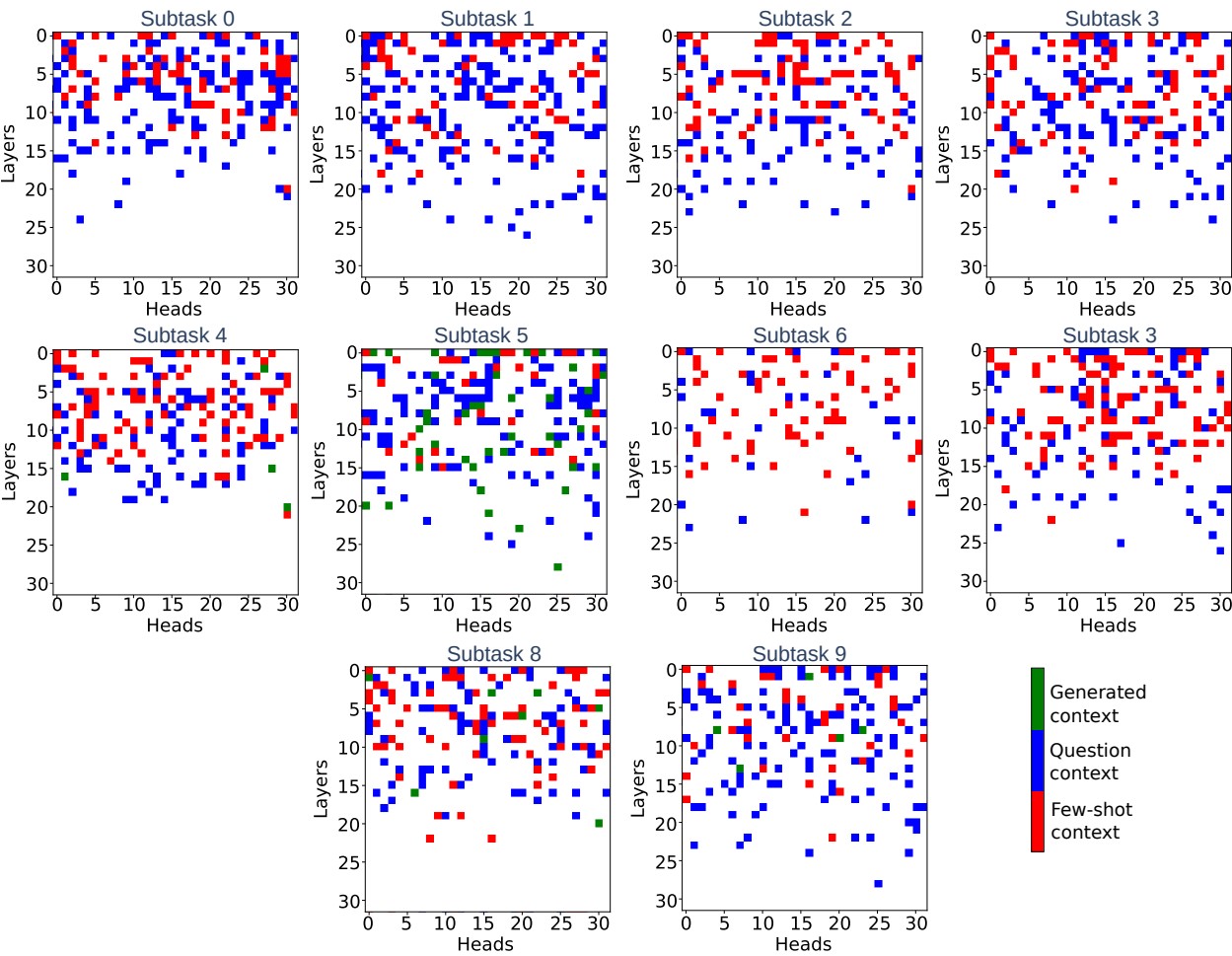

Figure 7: **Where do the answer-writing heads collect their answers from?** For each answer-writing head and each subtask, we show those attention heads that attend to residual streams corresponding to answer tokens. We demarcate heads that collect the answer tokens from the generated context (green), question context (blue), and few-shot context (red).

streams. We continue till one of the two conditions is met: (i) we reach a head in the first decoder block, or (ii) we reach a residual stream corresponding to the first token in the input token sequence. With such a procedure, we construct trees of attention heads rooted at the answer writing heads.

Now, let us consider the following example context plus generated token sequence: *Lempuses are tumpuses. Tumpuses are rompuses. Rompuses are blue. Max is lempus. True or false: Max is blue. Response: Let us think step by step. Max is lempus. Lempuses are tumpuses. Max is* with the desired prediction being *tumpus* (note that this input follows the few shot examples that are not shown here for brevity). We segregate the input context into three parts: few-shot context, question context, and generated context. We can see that the LLM can collect the answer *tumpus* from either generated or question context. Also, if the same token exists in the few-shot context, it can decide to collect information from there as well (however, that should not be ideal since the contextual role of that token would be different). We proceed to identify from the information-flow trees all those heads that attend to the streams corresponding to the answer token (in this example, *tumpus*). Note that there can be multiple other heads that might be attending to these same tokens; but only those heads that are elements of the tree rooted at answer writing heads are actually contributing to the flow of information from the answer tokens in the context of the output Figure 7 demonstrates these attention heads that attend to the residual streams corresponding to the answer tokens for different subtasks,

present in the generated context (green), question context (blue) and few-shot context (red). The following observations can be drawn from the figures:

1. **Coexitent pathways of answer generation**. Smaller models like GPT-2 on simpler tasks like IOI implements a unique neural algorithm (Wang et al., 2023). On the contrary, a larger model like Llama-2 exploits different parallel pathways of answer propagation while performing reasoning There are different heads that are directly connected to the answer writers, and they collect answer information from different places in the context.

2. **Different primary sources of answer for different subtasks.** In the initial stages of generation (i.e., subtasks 0, 1, 2, 3), the answer tokens are not present in the generated context. For subtasks 6 and 7, again, the answers need to be collected from the question index. For subtasks 4, 5, 8, and 9, we can observe the presence of heads that collect answer tokens from the context generated via earlier subtasks (in fact, subtask 5 has more such heads).

Given the fact that (i) there are multiple answer writing heads, and (ii) there are different heads that are directly connected to the answer writers, and they collect answer information from different places in the context, we can conclude that the different pathways implement different algorithms as well. In the few-shot examples, there are fictional entities that have been used in the question context as well, though in different contextual roles. The fact that there are attention heads collecting those entities as answers suggests that there are pathways prone to collecting similar information from the few-shot context. Although these pathways collect the same token as the answer, they actually deviate from the correct reasoning algorithm. However, the presence of such pathways decreases as the generation progresses from subtask 0 to 9. Note that this is different from few-shot examples providing the necessary pattern via in-context learning. Such patterns are more position-specific, i.e., while generating the answer for a certain subtask in the question, there is an overall increased attention provided to the same subtask tokens in the few-shot examples (see Figures 9, 10, 11, and 12 in Appendix). Finally, the pattern of answer collection from question context and generated context across different subtasks clearly demarcates the inductive reasoning subtasks (5 and 9) from the decision-making and copying. Tasks in the former category require pathways that collect answers from the earlier generated context, but the latter does not. Moreover, the decision-making step right before inductive reasoning can extract answers from the preceding step. This explains an albeit small number of heads that collect the answer from the generated context in subtasks 4 and 8. See Figures 13 and 14 for examples of such information flow towards subtasks 1 and 5, respectively.

## 7    Conclusion

**Our findings.** This work sought to disentangle the functional fabric of CoT reasoning in LLMs. Specifically, we explored Llama-2 7B in a few-shot CoT regime for solving multi-step reasoning problems on fictional ontologies of the PrOntoQA dataset. We observed that:

1. Despite different reasoning requirements across different stages of CoT generation, the functional components of the model remain almost the same. Different neural algorithms are implemented as compositions of induction circuit-like mechanisms.

2. Attention heads perform information movement between ontologically related (or negatively related) tokens. This information movement results in distinctly identifiable representations for such token pairs. Typically, this distinctive information movement starts from the very first layer and continues till the middle. While this phenomenon happens zero-shot, in-context examples exert pressure to quickly mix other task-specific information among tokens.

3. Multiple different neural pathways are deployed to compute the answer, that too in parallel. Different attention heads, albeit with different probabilistic certainty, write the answer token (for each CoT subtask) to the last residual stream.

4. These parallel answer generation pathways collect answers from different segments of the input. We found that while generating CoT, the model gathers answer tokens from the generated context, the question context, as well as the few-shot context. This provides a strong empirical answer to the open problem of whether LLMs actually use the context generated via CoT while answering questions.

5. We observe a functional rift at the very middle of the LLM (16th decoder block in case of Llama-2 7B), which marks a phase shift in the content of residual streams and the functionality of the attention heads. Prior to this rift, the model primarily assigns bigram associations memorized via pretraining; it drastically starts following the in-context prior to and after the rift. It is likely that this is directly related to the token-mixing along ontological relatedness that happens only prior to the rift. Similarly, answer-writing heads appear only after the rift. Attention heads that (wrongly) collect the answer token from the few-shot examples are also bounded by the prior half of the model.

**Limitations.** The design of this study imposes certain limitations in its scope. First and foremost, we analyze a very specific type of reasoning problem. While the fictional ontology with a restricted CoT template provides ease of analysis, free-form reasoning can introduce further complex dynamics not captured in this study. We also could not address the role of MLPs in reasoning. While existing literature points to their role as factual memory, it should be noted that the model also memorizes a diverse set of token-token associations pertaining to the structure of language and not just factual associations. Such associations are likely to play a decisive role in a grounded reasoning setup that our analysis ignores. Finally, we use a few-shot prompting regime to ensure that the model follows a specific structure of reasoning steps. With zero-shot CoT, more complex mechanisms are bound to emerge.

**Implications for future research.** These findings bear important ramifications towards the ongoing research around language modeling and interpretability. A natural extension to this work would be to incorporate pretraining memorization in terms of MLP blocks — precisely, whether the functional rift across layers bears a similar mechanism when the LLM starts mixing factual associations that have been stored within the MLP neurons. The existence of a parallel answer-generation process is extremely important for causal interventions on model behavior (Li et al., 2024): changing how a model should reason via up-(or down-) scaling certain neural pathways should take all the parallel pathways into account. Although real-world reasoning problems are far more dynamic than our strictly controlled setup of ontology-based reasoning, many of the results hold in the case of CoT in the wild. The existence of multiple neural pathways of answer generation is intrinsic to the model (although in real-world reasoning, such pathways will contain MLP blocks as a source of the answer, instead of the context only). The token-mixing mechanism that we uncovered remains true for any ontological relationship, not just fictional. Additionally, the more out-of-distribution (with respect to pretraining data) the reasoning problem becomes, the stronger the rift in terms of context abidance we will observe.

## Acknowledgments

The authors would like to acknowledge the financial support of DYSL-AI.

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

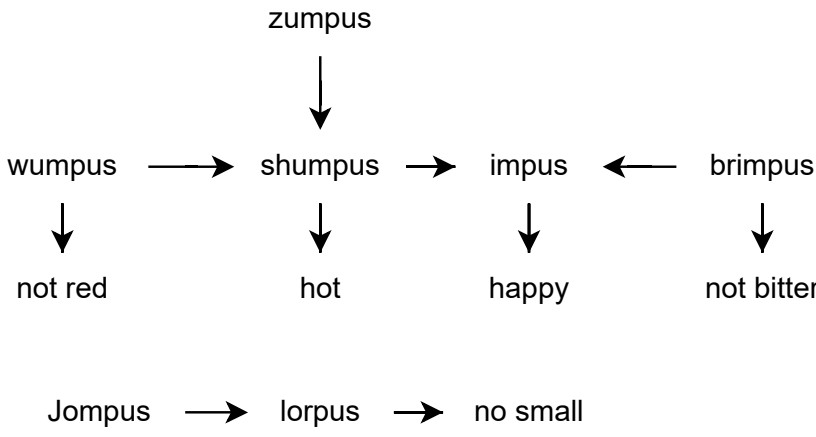

Figure 8: An example of fictional ontology in the PrOntoQA dataset.

## A  PrOntoQA question example

Following are two QA examples from fictional and false ontologies, respectively:

**Fictional ontology example:**
*Context:* Tumpus is bright. Lempus is tumpus. Max is lempus.
*Query:* True or false: Max is bright.
*Answer:* Max is lempus. Lempus is tumpus. Max is tumpus. Tumpus is bright. Max is bright. True
**False onotology example:**
*Context:* Carnivore is spicy. Vertebrates are carnivores. Rex is vertebrate.
*Query:* True or false: Rex is spicy.
*Answer:* Rex is vertebrate. Vertebrates are carnivores. Rex is carnivore. Carnivore is spicy. Rex is spicy. True

Figure 8 demonstrates an example ontology tree along with distractors. Here, *zumpus → shumpus* translates to *zumpuses are shumpuses.* From this example, one can construct the following multistep reasoning problem: *Wumpuses are shumpuses. Shumpuses are impuses. Max is a wumpus. Wumpuses are not red. Impuses are happy. Jompuses are lorpuses. True or False: Max is happy.* In this example, the statement *Jompuses are lorpuses* serves as a distractor as neither of the entities are part of the ontology relevant to the question.

## B  Knockout

To perform knockout average activation from false ontology the PrOntoQA dataset is used. For each subtask, false ontology activation is stored and mean activation knockout is performed. We provide step-by-step examples of subtask-wise generation over false ontology as follows:

*Context:* Carnivore is spicy. Vertebrates are carnivores. Rex is vertebrate.
*Query:* True or false: Rex is spicy.
*Answer*: "Rex is vertebrate. Vertebrates are carnivores. Rex is carnivore. Carnivore is spicy. Rex is spicy. True",
"prompt_0": "Carnivore is spicy. Vertebrates are carnivores. Rex is vertebrate. True or false: Rex is spicy. Let us think step by step.",
"prompt_1": "Carnivore is spicy. Vertebrates are carnivores. Rex is vertebrate. True or false: Rex is spicy. Let us think step by step. Rex is",
"prompt_2": "Carnivore is spicy. Vertebrates are carnivores. Rex is vertebrate. True or false: Rex is spicy. Let us think step by step. Rex is vertebrate.",
"prompt_3": "Carnivore is spicy. Vertebrates are carnivores. Rex is vertebrate. True or false: Rex is spicy.

Let us think step by step. Rex is vertebrate. Vertebrates are",

"prompt_4": "Carnivore is spicy. Vertebrates are carnivores. Rex is vertebrate. True or false: Rex is spicy. Let us think step by step. Rex is vertebrate. Vertebrates are carnivores.",

"prompt_5": "Carnivore is spicy. Vertebrates are carnivores. Rex is vertebrate. True or false: Rex is spicy. Let us think step by step. Rex is vertebrate. Vertebrates are carnivores. Rex is",

"prompt_6": "Carnivore is spicy. Vertebrates are carnivores. Rex is vertebrate. True or false: Rex is spicy. Let us think step by step. Rex is vertebrate. Vertebrates are carnivores. Rex is carnivore.",

"prompt_7": "Carnivore is spicy. Vertebrates are carnivores. Rex is vertebrate. True or false: Rex is spicy. Let us think step by step. Rex is vertebrate. Vertebrates are carnivores. Rex is carnivore. Carnivore is",

"prompt_8": "Carnivore is spicy. Vertebrates are carnivores. Rex is vertebrate. True or false: Rex is spicy. Let us think step by step. Rex is vertebrate. Vertebrates are carnivores. Rex is carnivore. Carnivore is spicy.",

"prompt_9": "Carnivore is spicy. Vertebrates are carnivores. Rex is vertebrate. True or false: Rex is spicy. Let us think step by step. Rex is vertebrate. Vertebrates are carnivores. Rex is carnivore. Carnivore is spicy. Rex is spicy.",

## C    Task specific head identification

### C.1    Decision-making heads

To identify the heads that actively participate in decision-making subtasks, we incorporate activation patching on individual heads over decision-making subtasks.

Let $S_{\text{Decision}}$ denote the input token sequence for a particular decision-making subtask with $s_{\text{ans}}$ being the token corresponding to the correct answer. Also, let $P_{\text{org}}$ denote the output token probability distribution of the full model corresponding to $S_{\text{Decision}}$ as input, i.e.,

$$P_{\text{org}} = \text{SoftMax}\left(\text{LM}(\boldsymbol{x}_{\text{logit}}|S_{\text{Decision}})\right)$$

Here we omit the token position superscript over $\boldsymbol{x}_{\text{logit}}$ for brevity since we are taking the output corresponding to the last input token. We store the activations $\boldsymbol{y}_{j,k}$ corresponding to each head $h_{j,k}$.

Next, we corrupt the input corresponding to the last token in $S_{\text{Decision}}$, which is equivalent to knocking off the set of all heads, $\mathcal{H}_{\text{full}}$. We record the corresponding output token probability as,

$$P_{\text{corrupt}} = \text{SoftMax}\left(\text{LM}_{\mathcal{H}_{\text{full}}}\left(\boldsymbol{x}_{\text{logit}}|S_{\text{Decision}}\right)\right)$$

Finally, for each head $h_{j,k}$, we restore its corresponding original output $\boldsymbol{y}_{j,k}$ and record the output token probability distribution as

$$P_{\text{patched}}^{j,k} = \text{SoftMax}\left(\text{LM}_{\mathcal{H}_{\text{full}}\setminus\{h_{j,k}\}}\left(\boldsymbol{x}_{\text{logit}}|S_{\text{Decision}}\right)\right)$$

Then, we compute the importance score of head $h_{j,k}$ as

$$\mu_{\text{Decision}}(h_{j,k}) = \frac{P_{\text{org}}(x=s_{\text{ans}}) - P_{\text{corrupt}}(x=s_{\text{ans}})}{P_{\text{org}}(x=s_{\text{ans}}) - P_{\text{patched}}(x=s_{\text{ans}})}$$

### C.2    Copy heads

Copying subtasks follows decision-making subtasks immediately. Once the head entity of a statement is decided, copying requires moving the tail entity corresponding to that statement in the input context to the output (e.g., if there is a statement *Rompus is grimpus* in the input context and decision-making heads have decided to output *Rompus*, then copy subtask requires moving *grimpus* to the output). For this, we simply look into the attention probability assigned by each head $h_{j,k}$ to the token to be copied as source (i.e., key),

denoted as

$$\mu_{\text{Copy}}(h_{j,k}) = \frac{\exp\left((\boldsymbol{W}_Q^{j,k}\boldsymbol{x}_j^{\text{end}})^\top (\boldsymbol{W}_K^{j,k}\boldsymbol{x}_j^{\text{ans}})\right)}{\sum_i \exp\left((\boldsymbol{W}_Q^{j,k}\boldsymbol{x}_j^{\text{end}})^\top (\boldsymbol{W}_K^{j,k}\boldsymbol{x}_j^i)\right)}$$

where $\boldsymbol{x}_j^{\text{end}}, \boldsymbol{x}_j^{\text{ans}}$ denote the residual streams corresponding to the last token and the answer token, respectively, at $j$-th decoder block.

### C.3  Inductive reasoning heads

Let the input token sequence for the inductive reasoning subtask be denoted as $S_{\text{ind}}$, which is of the form [A] is [B]. [B] is [C]. [A] is, and let $s_{\text{ans}}$ denote the answer token (which is [C] in this example). We represent the first and second occurrences of each token [A] and [B] as $A_1, A_2$ and $B_1, B_2$, respectively. For a given $S_{\text{Induction}}$, we can perform activation patching on three different residual streams, $\boldsymbol{x}_j^{B_1}$, $\boldsymbol{x}_j^{B_2}$, and $\boldsymbol{x}_j^C$ to identify the responsible heads. Let the original token probability distribution be

$$P_{\text{org}} = \text{SoftMax}\left(\text{LM}(\boldsymbol{x}_{\text{logit}}|S_{\text{Induction}})\right)$$

Next, for each $l \in \{B_1, B_2, C\}$, we compute the corrupted forward pass as follows:

$$P_{\text{corrupt},l} = \text{SoftMax}\left(\text{LM}_{\mathcal{H}_{\text{full}}}^l(\boldsymbol{x}_{\text{logit}}|S_{\text{Induction}})\right)$$

followed by patching original activation to the head $h_{j,k}$ as

$$P_{\text{patched},l}^{j,k} = \text{SoftMax}\left(\text{LM}_{\mathcal{H}_{\text{full}}\setminus\{h_{j,k}\}}^l(\boldsymbol{x}_{\text{logit}}|S_{\text{Induction}})\right)$$

Then, the inductive reasoning head importance is computed as:

$$\mu_{\text{Induction}}(h_{j,k}) = \text{Mean}_l \frac{D_{\text{ref}}^l - D_{\text{patched}}^l}{D_{\text{ref}}^l}$$

where $D_{\text{ref}}^l$ is the KL divergence between $P_{\text{org}}$ and $P_{\text{corrupt},l}$, and $D_{\text{patched}}^l$ is the KL divergence between $P_{\text{org}}$ and $P_{\text{corrupt},l}$.

### C.4  Performance of inductive reasoning heads for each subtask

For each subtask 0-9, we group the attention heads based on their respective $\mu_{\text{Induction}}(h_{j,k})$. We knockout heads within a certain threshold head importance range $(\mu_{\text{min}}, \mu_{\text{max}})$ by knockout and record the accuracy. In Table 1, we report the attention head statistics that we used in the analysis for context abidance and answer-writing pathways. Note that we sought to keep the relative accuracy (i.e., the fraction of correct prediction by the ablated model over the correct predictions of the full model) close to 0.9.

## D  Probing for token mixing

Towards probing ontological relatedness in residual streams (see Section 5), first, we need to extract the positive, negative and unrelated entities from the sentence. PrOntoQA provides us with a tree structure for each sentence, and with the help of the structure, we can extract the pairs. Consider the following example.

Sentence: *Each shumpus is a zumpus. Shumpuses are wumpuses. Every shumpus is hot. Impuses are shumpuses. Each impus is a brimpus. Each impus is happy. Wumpuses are not red. Brimpuses are not bitter. Lorpuses are jompuses. Each yumpus is not hot. Lorpuses are not small. Max is an impus. Max is a lorpus.*

Here all immediate entities will be termed as posive pairs, such as *shumpus <> impus, zumpus <> shumpus, impus<> happy*, etc. Similarly, following are a few examples of negatively related entities: *wumpus <>*

| Subtask index | Accuracy | Heads removed | Threshold range |
|---|---|---|---|
| 0 | 1 | 475 | 0.30487806 - 0.31463414 |
| 1 | 0.93 | 554 | 0.30243903,0.31707317 |
| 2 | 0.96 | 617 | 0.3 - 0.3195122 |
| 3 | 1 | 617 | 0.3 - 0.3195122 |
| 4 | 0.99 | 554 | 0.30243903 - 0.31707317 |
| 5 | 0.93 | 475 | 0.30487806 - 0.31463414 |
| 6 | 0.94 | 475 | 0.30487806 - 0.31463414 |
| 7 | 0.88 | 617 | 0.3 - 0.3195122 |
| 8 | 0.95 | 617 | 0.3 - 0.3195122 |
| 9 | 0.91 | 663 | 0.297561 - 0.3219512 |

Table 1: Statistics of subtask-wise attention head removal according to their importance in inductive reasoning.

*red*, *brimpus <> bitter*, etc. The distractor entities (jompus, lorpus and small) serves as seed for unrelated entities; any two entities that belong two disjoint ontologies are taken as unrelated.

During training and testing, we also make sure that there are no overlap between entities. During training, we use entities such as "wumpus", "yumpus", "zumpus", "dumpus" and during testing : "zonkify", "quiblitz","flimjam", "zizzlewump", "snickerblat". Following are the implementation details for the probing classifier:

- Total training pairs: 28392; Total testing pairs: 9204. All three types of pairs (positively and negatively related and unrelated) are present in equal proportion in the training and testing data.

- 4-layer MLP model, 4096 * 2 -> 128 -> 64 -> 32 -> 3. With ReLU in between each Linear layer.

- Learning rate: 0.00005

- Number of epochs: 120.

# E   Prompts used and model performance

We use 6-shot examples of CoT for generation in all the experiments.:

### Input:
Gorpus is twimpus. Alex is rompus. Rompus is gorpus. Gorpus is small. Rompus is mean. True or false: Alex is small. Let us think step by step.
### Response:
Alex is rompus. Rompus is gorpus. Alex is gorpus. Gorpus is small. Alex is small. True

### Input:
Gorpuses are discordant. Max is zumpus. Zumpus is shampor. Zumpus is gorpus. Gorpus is earthy. True or false: Max is small. Let us think step by step.
### Response:
Max is zumpus. Zumpus is gorpus. Max is gorpus. Gorpuses are discordant. Max is discordant. False

### Input:
Borpin are wumpus. Wumpuses are angry. Wumpus is jempor. Sally is lempus. Lempus is wumpus. True or false: Sally is floral. Let us think step by step.
### Response:
Sally is lempus. Lempus is wumpus. Sally is wumpus. Wumpuses are angry. Sally is angry. False

### Input:
Gorpus is jelgit. Yumpuses are loud. Gorpus is yumpus. Yumpus is orange. Rex is gorpus. True or false:

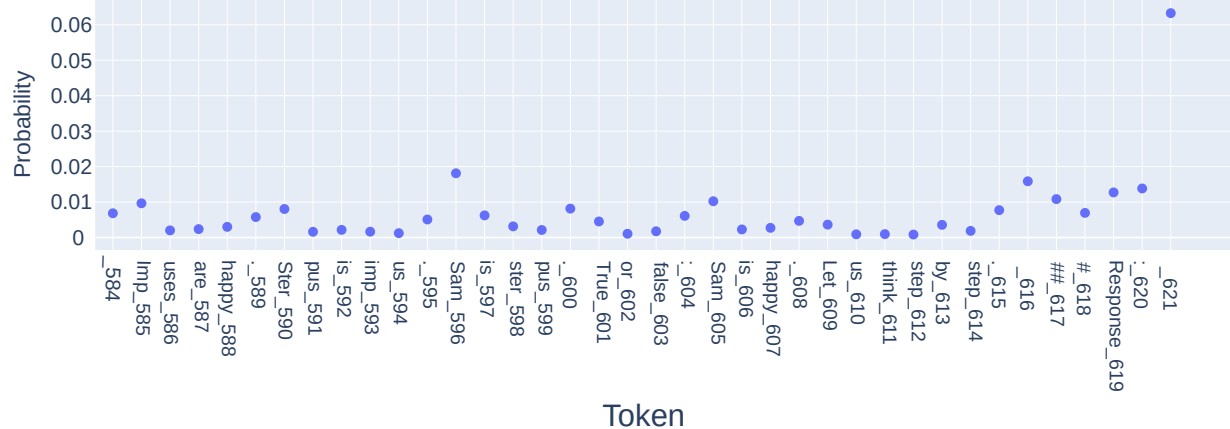

Figure 9: Average attention probability distribution (across all heads) over question context in an example subtask 0. Here, the model needs to predict the token *Sam*.

Rex is loud. Let us think step by step.
### Response:
Rex is gorpus. Gorpus is yumpus. Rex is yumpus. Yumpuses are loud. Rex is loud. True

### Input:
Lempus is tumpus. Max is lempus. Tumpus is fruity. Tumpus is bright. Lempus is dropant. True or false: Max is bright. Let us think step by step.
### Response:
Max is lempus. Lempus is tumpus. Max is tumpus. Tumpus is bright. Max is bright. True

### Input:
Sterpuses are dull. Impus is medium. Impuses are sterpuses. Wren is impus. Sterpus is daumpin. True or false: Wren is melodic. Let us think step by step.
### Response:
Wren is impus. Impuses are sterpuses. Wren is sterpus. Sterpuses are dull. Wren is dull. False

To ensure that the model does not learn any positional pattern from the few-shot example, we jumble the statements in the Input context and add two extra distractors in the statement.

Following are the performance of Llama-2-7B on two subdomains of PrOntoQA — (i) accuracy on false ontology: 52.5, (ii) accuracy on fictional ontology: 35.9.

Accuracy is calculated by a complete match of all the intermediate reasoning subtasks and the final True/False statement. We make the following modification to the PrOntoQA dataset: (i) removal of articles like *a* and *an* from statements, (ii) removal of *every* and *each* from statements, (iii) replacing negative entities with incorrect entities (e.g., *Max is not cold → Max is dog*).

# F   Attention probability

We take the average of attention probability at the last residual stream of all the heads. Figure 9 shows the attention probability over the input context tokens, and Figures 10, 11, and 12 show attention probability over the 6-shot examples used. This example is for subtask-0, and it can be seen that there are spikes in attention probability score for tokens at the same position as subtask 0, i.e., token just after "Response:". Similar patterns are observed for all the subtasks. Furthermore, the absolute value of attention to the task-specific tokens in the few-shot context increases as we move from the 1st to 6th example.

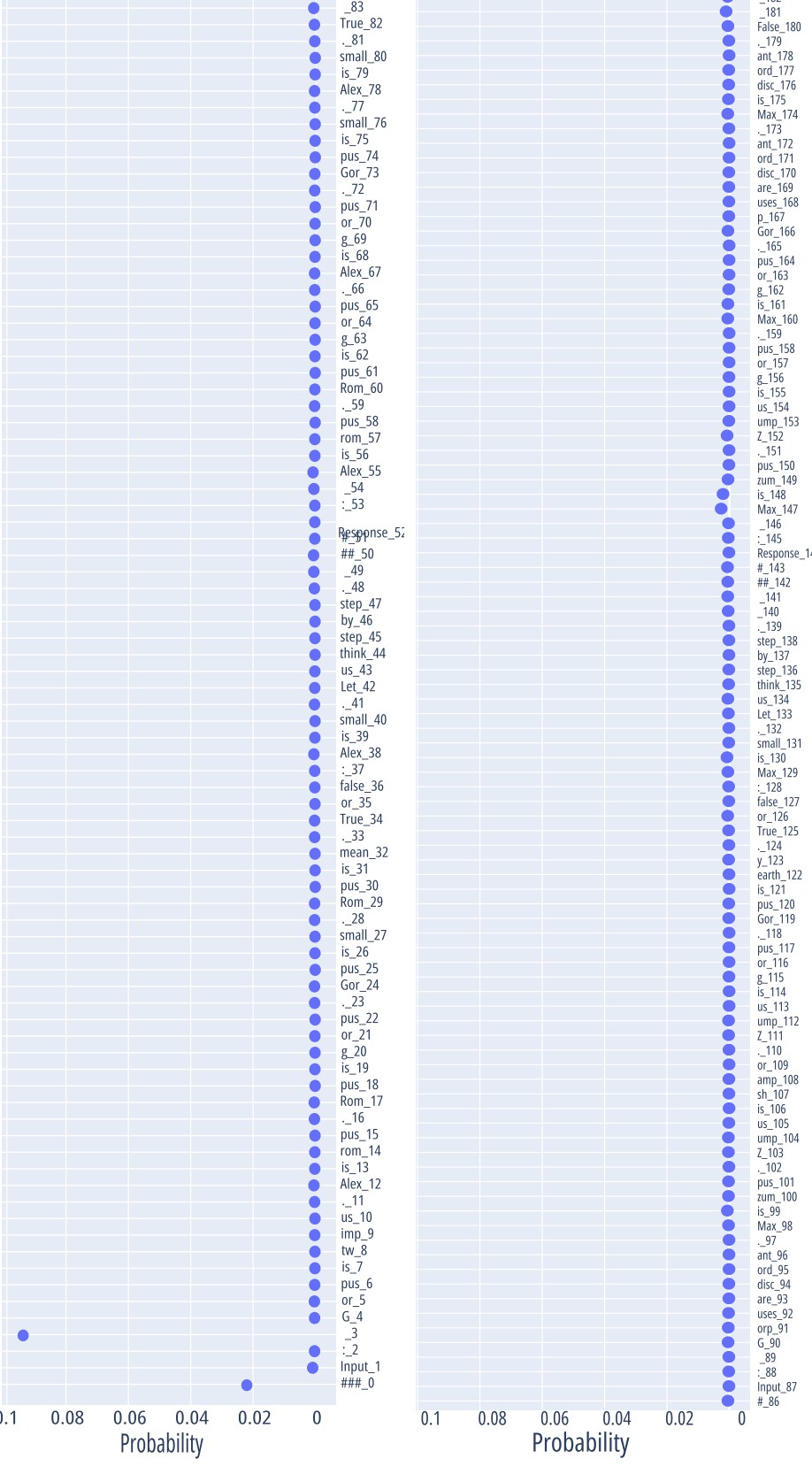

Figure 10: **Attention probability distribution over few-shot examples**. In this plot, we show the average attention scores over tokens corresponding to the first (left) and second (right) examples (for the rest of the examples, see Figure 11 and 12) among the 6-shot context corresponding to the question in Figure 9. Here, the model is at subtask-0 and needs to predict *Sam*. A slight peak in attention probability at the token corresponding to subtask-0 in the second example (*Max* in this case) is observable.

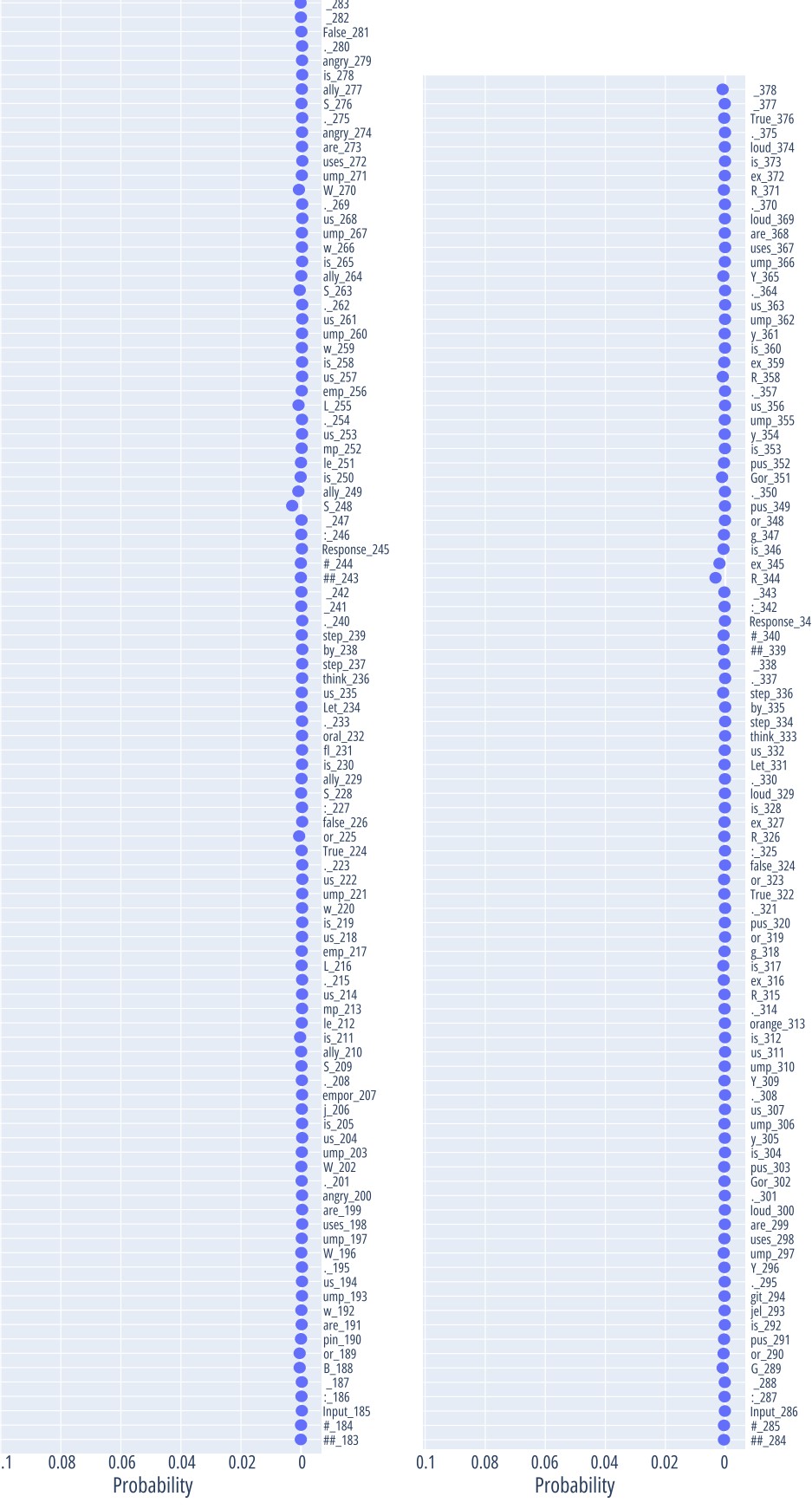

Figure 11: **Attention probability distribution over few-shot examples** (continued after Figure 10). In this plot, we show the average attention scores over tokens corresponding to the third (left) and fourth (right) examples among the few-shot context corresponding to the question in Figure 9. Here, the model is at subtask-0 and needs to predict *Sam*. Small peaks in attention probability at tokens corresponding to subtask-0 in the third (*S* from *Sally*) and fourth (*R* from *Rex*) example are observable. For the rest of the examples, see Figure 12

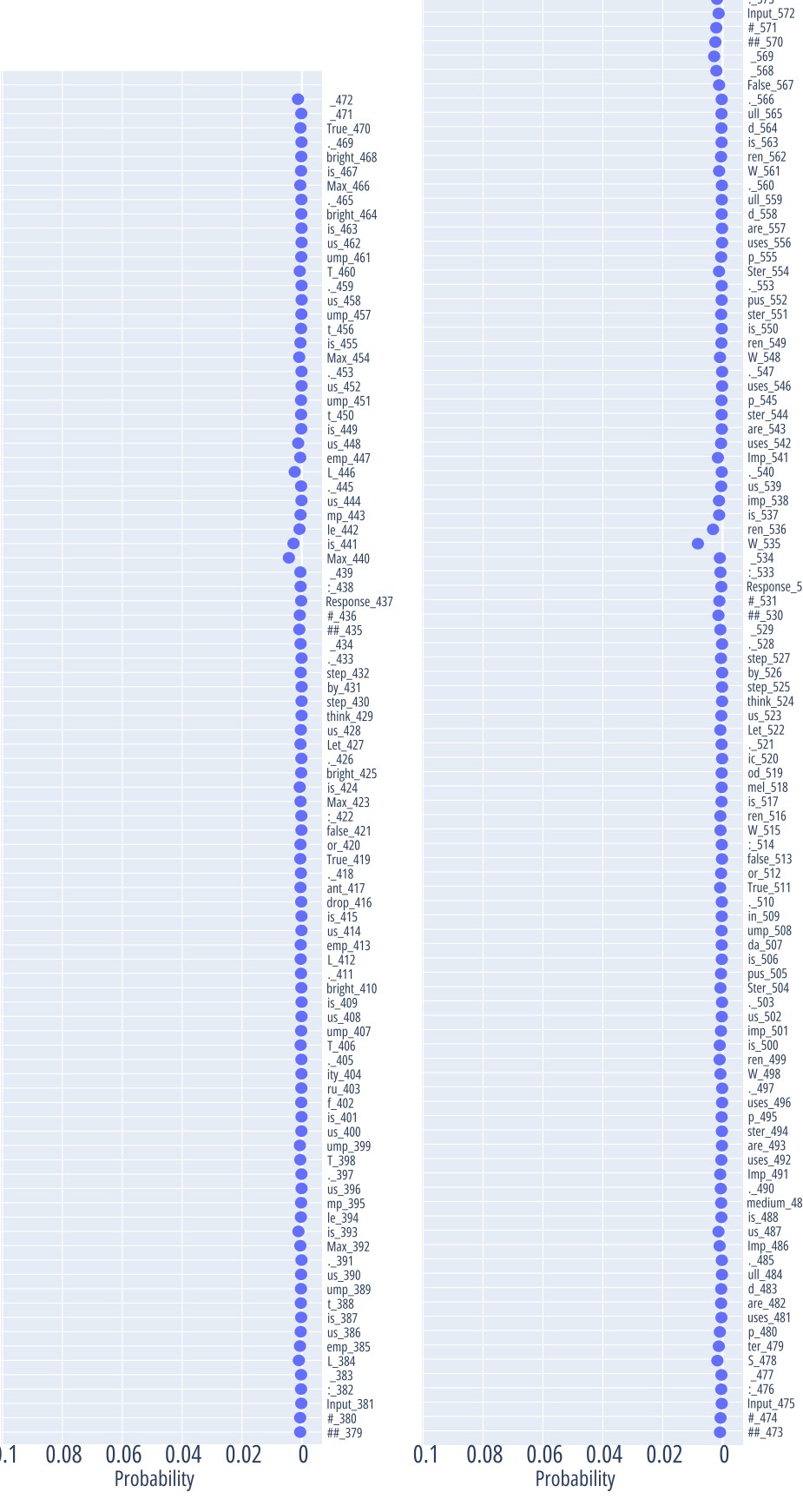

Figure 12: **Attention probability distribution over few-shot examples** (continued after Figure 10 and 11). In this plot, we show the average attention scores over tokens corresponding to the fifth (left) and sixth (right) examples among the few-shot context corresponding to the question in Figure 9. Here, the model is at subtask-0 and needs to predict *Sam*. Small peaks in attention probability at tokens corresponding to subtask-0 in the fifth (*Max*) and sixth (*W* from *Wren*) example are observable.

## G    Information flow

Assume a given head $h_{j,k}$ that writes some information to the residual stream corresponding to the token $s_i$ via output $\boldsymbol{y}_{j,k}^i$. Furthermore, let the maximum attention weight allocated by $h_{j,k}$ be placed on the residual stream corresponding to token $s_l$. Then, we represent the head with the triplet $(s_l, s_i, s_{\text{info}} = \arg\max \boldsymbol{U}\boldsymbol{y}_{j,k}^i)$ denoting the source residual stream (source of information written by the head), target residual stream (target of information written by the head), and the information that is being written.

We start with the answer writing heads, $h_{j,k}^{\text{Ans}}$. For $(s_l, s_i, s_{\text{info}})$ associated with each $h_{j,k}^{\text{Ans}}$, we identify the information content present within the source residual stream as $\arg\max \boldsymbol{U}\boldsymbol{x}_{j-1}^l$ (i.e., the information present within the source residual stream right before the answer writer attended to it). Then, we look for all the heads in decoder blocks $< j >$ responsible for writing that information into that residual stream. This continues till we reach a head that is in the starting decoder block. Figures 13 depict information flow extracted for a single example question for subtask 1. The tokenized input is as follows:

```
_0 _1 _2 _3 _4 _5 ###_6 Input_7 :_8 <0x0A>_9 G_10 or_11 pus_12 is_13
tw_14 imp_15 us_16 ._17 Alex_18 is_19 rom_20 pus_21 ._22 Rom_23 pus_24 is_25 g_26 or_27
pus_28 ._29 Gor_30 pus_31 is_32 small_33 ._34 Rom_35 pus_36 is_37 mean_38 ._39 True_40
or_41 false_42 :_43 Alex_44 is_45 small_46 ._47 Let_48 us_49 think_50 step_51 by_52
step_53 ._54 <0x0A>_55 ##_56 #_57 Response_58 :_59 <0x0A>_60 Alex_61 is_62 rom_63 pus_64
._65 Rom_66 pus_67 is_68 g_69 or_70 pus_71 ._72 Alex_73 is_74 g_75 or_76 pus_77 ._78
Gor_79 pus_80 is_81 small_82 ._83 Alex_84 is_85 small_86 ._87 True_88 <0x0A>_89 <0x0A>_90
##_91 #_92 Input_93 :_94 <0x0A>_95 G_96 orp_97 uses_98 are_99 disc_100 ord_101 ant_102
._103 Max_104 is_105 zum_106 pus_107 ._108 Z_109 ump_110 us_111 is_112 sh_113 amp_114
or_115 ._116 Z_117 ump_118 us_119 is_120 g_121 or_122 pus_123 ._124 Gor_125 pus_126
is_127 earth_128 y_129 ._130 True_131 or_132 false_133 :_134 Max_135 is_136 small_137
._138 Let_139 us_140 think_141 step_142 by_143 step_144 ._145 _146 <0x0A>_147 ##_148
#_149 Response_150 :_151 <0x0A>_152 Max_153 is_154 zum_155 pus_156 ._157 Z_158 ump_159
us_160 is_161 g_162 or_163 pus_164 ._165 Max_166 is_167 g_168 or_169 pus_170 ._171
Gor_172 p_173 uses_174 are_175 disc_176 ord_177 ant_178 ._179 Max_180 is_181 disc_182
ord_183 ant_184 ._185 False_186 <0x0A>_187 <0x0A>_188 ##_189 #_190 Input_191 :_192
<0x0A>_193 B_194 or_195 pin_196 are_197 w_198 ump_199 us_200 ._201 W_202 ump_203 uses_204
are_205 angry_206 ._207 W_208 ump_209 us_210 is_211 j_212 empor_213 ._214 S_215 ally_216
is_217 le_218 mp_219 us_220 ._221 L_222 emp_223 us_224 is_225 w_226 ump_227 us_228 ._229
True_230 or_231 false_232 :_233 S_234 ally_235 is_236 fl_237 oral_238 ._239 Let_240
us_241 think_242 step_243 by_244 step_245 ._246 _247 <0x0A>_248 ##_249 #_250 Response_251
:_252 <0x0A>_253 S_254 ally_255 is_256 le_257 mp_258 us_259 ._260 L_261 emp_262 us_263
is_264 w_265 ump_266 us_267 ._268 S_269 ally_270 is_271 w_272 ump_273 us_274 ._275 W_276
ump_277 uses_278 are_279 angry_280 ._281 S_282 ally_283 is_284 angry_285 ._286 False_287
<0x0A>_288 <0x0A>_289 ##_290 #_291 Input_292 :_293 <0x0A>_294 G_295 or_296 pus_297
is_298 jel_299 git_300 ._301 Y_302 ump_303 uses_304 are_305 loud_306 ._307 Gor_308
pus_309 is_310 y_311 ump_312 us_313 ._314 Y_315 ump_316 us_317 is_318 orange_319 ._320
R_321 ex_322 is_323 g_324 or_325 pus_326 ._327 True_328 or_329 false_330 :_331 R_332
ex_333 is_334 loud_335 ._336 Let_337 us_338 think_339 step_340 by_341 step_342 ._343
<0x0A>_344 ##_345 #_346 Response_347 :_348 <0x0A>_349 R_350 ex_351 is_352 g_353 or_354
pus_355 ._356 Gor_357 pus_358 is_359 y_360 ump_361 us_362 ._363 R_364 ex_365 is_366 y_367
ump_368 us_369 ._370 Y_371 ump_372 uses_373 are_374 loud_375 ._376 R_377 ex_378 is_379
loud_380 ._381 True_382 <0x0A>_383 <0x0A>_384 ##_385 #_386 Input_387 :_388 <0x0A>_389
L_390 emp_391 us_392 is_393 t_394 ump_395 us_396 ._397 Max_398 is_399 le_400 mp_401
us_402 ._403 T_404 ump_405 us_406 is_407 f_408 ru_409 ity_410 ._411 T_412 ump_413 us_414
is_415 bright_416 ._417 L_418 emp_419 us_420 is_421 drop_422 ant_423 ._424 True_425
or_426 false_427 :_428 Max_429 is_430 bright_431 ._432 Let_433 us_434 think_435 step_436
by_437 step_438 ._439 <0x0A>_440 ##_441 #_442 Response_443 :_444 <0x0A>_445 Max_446
is_447 le_448 mp_449 us_450 ._451 L_452 emp_453 us_454 is_455 t_456 ump_457 us_458 ._459
Max_460 is_461 t_462 ump_463 us_464 ._465 T_466 ump_467 us_468 is_469 bright_470 ._471
```

Max_472 is_473 bright_474 ._475 True_476 <0x0A>_477 <0x0A>_478 ##_479 #_480 Input_481
:_482 <0x0A>_483 S_484 ter_485 p_486 uses_487 are_488 d_489 ull_490 ._491 Imp_492 us_493
is_494 medium_495 ._496 Imp_497 uses_498 are_499 ster_500 p_501 uses_502 ._503 W_504
ren_505 is_506 imp_507 us_508 ._509 Ster_510 pus_511 is_512 da_513 ump_514 in_515 ._516
True_517 or_518 false_519 :_520 W_521 ren_522 is_523 mel_524 od_525 ic_526 ._527 Let_528
us_529 think_530 step_531 by_532 step_533 ._534 <0x0A>_535 ##_536 #_537 Response_538
:_539 <0x0A>_540 W_541 ren_542 is_543 imp_544 us_545 ._546 Imp_547 uses_548 are_549
ster_550 p_551 uses_552 ._553 W_554 ren_555 is_556 ster_557 pus_558 ._559 Ster_560 p_561
uses_562 are_563 d_564 ull_565 ._566 W_567 ren_568 is_569 d_570 ull_571 ._572 False_573
<0x0A>_574 <0x0A>_575 ##_576 #_577 Input_578 :_579 <0x0A>_580 J_581 om_582 pus_583 is_584
wooden_585 ._586 Imp_587 us_588 is_589 j_590 om_591 pus_592 ._593 St_594 ella_595 is_596
imp_597 us_598 ._599 True_600 or_601 false_602 :_603 St_604 ella_605 is_606 wooden_607
._608 Let_609 us_610 think_611 step_612 by_613 step_614 ._615 <0x0A>_616 ##_617 #_618
Response_619 :_620 <0x0A>_621 St_622 ella_623 is_624

Figure 14 depict information flow extracted for a single example question for subtask 5. The tokenized
input is as follows:

_0 _1 _2 _3 _4 _5 _6 ###_7 Input_8 :_9 <0x0A>_10 G_11 or_12 pus_13
is_14 tw_15 imp_16 us_17 ._18 Alex_19 is_20 rom_21 pus_22 ._23 Rom_24 pus_25 is_26 g_27
or_28 pus_29 ._30 Gor_31 pus_32 is_33 small_34 ._35 Rom_36 pus_37 is_38 mean_39 ._40
True_41 or_42 false_43 :_44 Alex_45 is_46 small_47 ._48 Let_49 us_50 think_51 step_52
by_53 step_54 ._55 <0x0A>_56 ##_57 #_58 Response_59 :_60 <0x0A>_61 Alex_62 is_63 rom_64
pus_65 ._66 Rom_67 pus_68 is_69 g_70 or_71 pus_72 ._73 Alex_74 is_75 g_76 or_77 pus_78
._79 Gor_80 pus_81 is_82 small_83 ._84 Alex_85 is_86 small_87 ._88 True_89 <0x0A>_90
<0x0A>_91 ##_92 #_93 Input_94 :_95 <0x0A>_96 G_97 orp_98 uses_99 are_100 disc_101 ord_102
ant_103 ._104 Max_105 is_106 zum_107 pus_108 ._109 Z_110 ump_111 us_112 is_113 sh_114
amp_115 or_116 ._117 Z_118 ump_119 us_120 is_121 g_122 or_123 pus_124 ._125 Gor_126
pus_127 is_128 earth_129 y_130 ._131 True_132 or_133 false_134 :_135 Max_136 is_137
small_138 ._139 Let_140 us_141 think_142 step_143 by_144 step_145 ._146 _147 <0x0A>_148
##_149 #_150 Response_151 :_152 <0x0A>_153 Max_154 is_155 zum_156 pus_157 ._158 Z_159
ump_160 us_161 is_162 g_163 or_164 pus_165 ._166 Max_167 is_168 g_169 or_170 pus_171
._172 Gor_173 p_174 uses_175 are_176 disc_177 ord_178 ant_179 ._180 Max_181 is_182
disc_183 ord_184 ant_185 ._186 False_187 <0x0A>_188 <0x0A>_189 ##_190 #_191 Input_192
:_193 <0x0A>_194 B_195 or_196 pin_197 are_198 w_199 ump_200 us_201 ._202 W_203 ump_204
uses_205 are_206 angry_207 ._208 W_209 ump_210 us_211 is_212 j_213 empor_214 ._215 S_216
ally_217 is_218 le_219 mp_220 us_221 ._222 L_223 emp_224 us_225 is_226 w_227 ump_228
us_229 ._230 True_231 or_232 false_233 :_234 S_235 ally_236 is_237 fl_238 oral_239
._240 Let_241 us_242 think_243 step_244 by_245 step_246 ._247 _248 <0x0A>_249 ##_250
#_251 Response_252 :_253 <0x0A>_254 S_255 ally_256 is_257 le_258 mp_259 us_260 ._261
L_262 emp_263 us_264 is_265 w_266 ump_267 us_268 ._269 S_270 ally_271 is_272 w_273
ump_274 us_275 ._276 W_277 ump_278 uses_279 are_280 angry_281 ._282 S_283 ally_284 is_285
angry_286 ._287 False_288 <0x0A>_289 <0x0A>_290 ##_291 #_292 Input_293 :_294 <0x0A>_295
G_296 or_297 pus_298 is_299 jel_300 git_301 ._302 Y_303 ump_304 uses_305 are_306 loud_307
._308 Gor_309 pus_310 is_311 y_312 ump_313 us_314 ._315 Y_316 ump_317 us_318 is_319
orange_320 ._321 R_322 ex_323 is_324 g_325 or_326 pus_327 ._328 True_329 or_330 false_331
:_332 R_333 ex_334 is_335 loud_336 ._337 Let_338 us_339 think_340 step_341 by_342
step_343 ._344 <0x0A>_345 ##_346 #_347 Response_348 :_349 <0x0A>_350 R_351 ex_352 is_353
g_354 or_355 pus_356 ._357 Gor_358 pus_359 is_360 y_361 ump_362 us_363 ._364 R_365 ex_366
is_367 y_368 ump_369 us_370 ._371 Y_372 ump_373 uses_374 are_375 loud_376 ._377 R_378
ex_379 is_380 loud_381 ._382 True_383 <0x0A>_384 <0x0A>_385 ##_386 #_387 Input_388
:_389 <0x0A>_390 L_391 emp_392 us_393 is_394 t_395 ump_396 us_397 ._398 Max_399 is_400
le_401 mp_402 us_403 ._404 T_405 ump_406 us_407 is_408 f_409 ru_410 ity_411 ._412 T_413
ump_414 us_415 is_416 bright_417 ._418 L_419 emp_420 us_421 is_422 drop_423 ant_424 ._425
True_426 or_427 false_428 :_429 Max_430 is_431 bright_432 ._433 Let_434 us_435 think_436

step_437 by_438 step_439 ._440 <0x0A>_441 ##_442 #_443 Response_444 :_445 <0x0A>_446
Max_447 is_448 le_449 mp_450 us_451 ._452 L_453 emp_454 us_455 is_456 t_457 ump_458
us_459 ._460 Max_461 is_462 t_463 ump_464 us_465 ._466 T_467 ump_468 us_469 is_470
bright_471 ._472 Max_473 is_474 bright_475 ._476 True_477 <0x0A>_478 <0x0A>_479 ##_480
#_481 Input_482 :_483 <0x0A>_484 S_485 ter_486 p_487 uses_488 are_489 d_490 ull_491
._492 Imp_493 us_494 is_495 medium_496 ._497 Imp_498 uses_499 are_500 ster_501 p_502
uses_503 ._504 W_505 ren_506 is_507 imp_508 us_509 ._510 Ster_511 pus_512 is_513 da_514
ump_515 in_516 ._517 True_518 or_519 false_520 :_521 W_522 ren_523 is_524 mel_525 od_526
ic_527 ._528 Let_529 us_530 think_531 step_532 by_533 step_534 ._535 <0x0A>_536 ##_537
#_538 Response_539 :_540 <0x0A>_541 W_542 ren_543 is_544 imp_545 us_546 ._547 Imp_548
uses_549 are_550 ster_551 p_552 uses_553 ._554 W_555 ren_556 is_557 ster_558 pus_559
._560 Ster_561 p_562 uses_563 are_564 d_565 ull_566 ._567 W_568 ren_569 is_570 d_571
ull_572 ._573 False_574 <0x0A>_575 <0x0A>_576 ##_577 #_578 Input_579 :_580 <0x0A>_581
J_582 om_583 pus_584 is_585 wooden_586 ._587 Imp_588 us_589 is_590 j_591 om_592 pus_593
._594 St_595 ella_596 is_597 imp_598 us_599 ._600 True_601 or_602 false_603 :_604 St_605
ella_606 is_607 wooden_608 ._609 Let_610 us_611 think_612 step_613 by_614 step_615 ._616
<0x0A>_617 ##_618 #_619 Response_620 :_621 <0x0A>_622 St_623 ella_624 is_625 imp_626
us_627 ._628 Imp_629 us_630 is_631 j_632 om_633 pus_634 ._635 St_636 ella_637 is_638

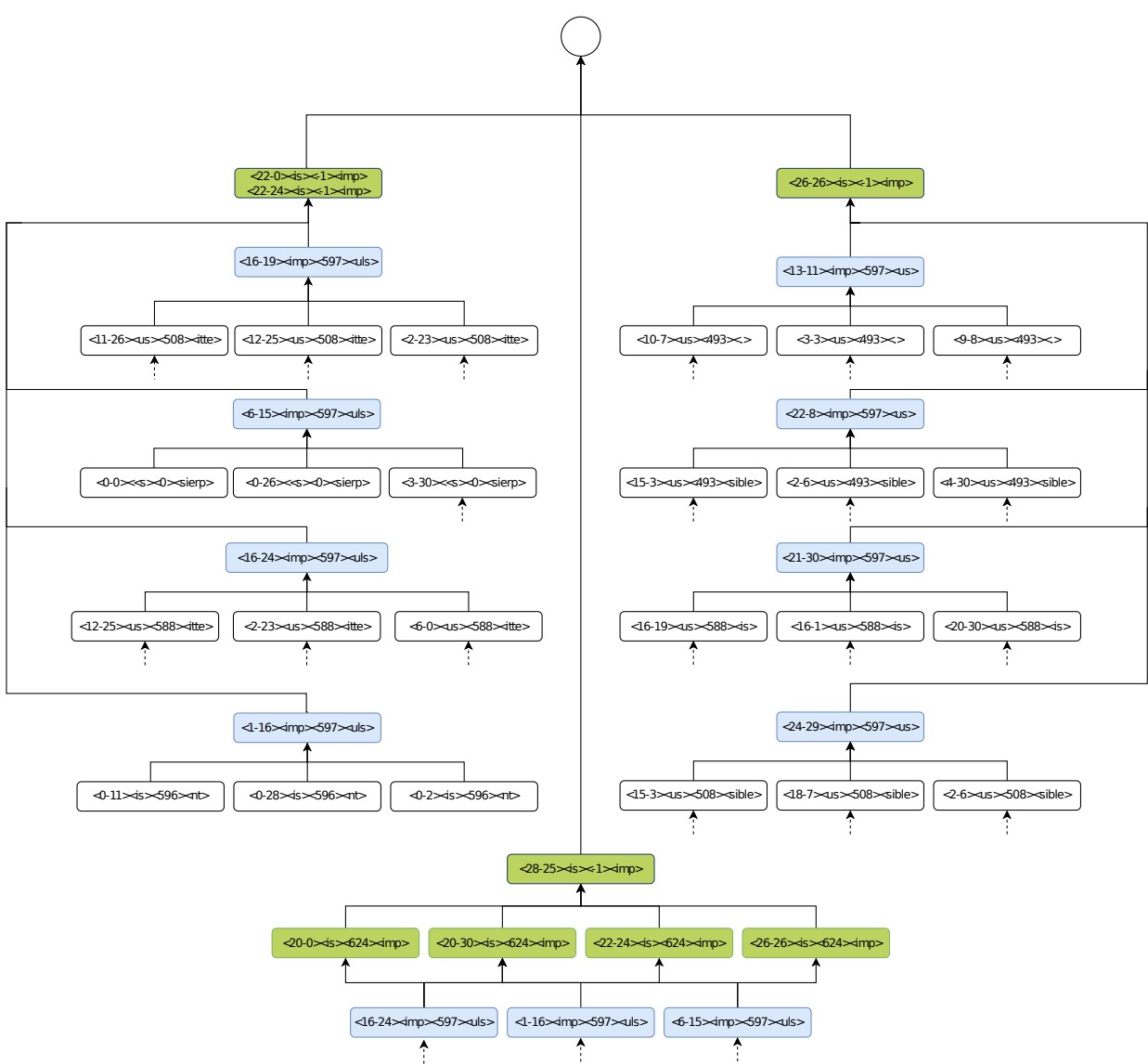

Figure 13: **Infomation flow through attention heads towards answer-writing heads in Llama-2 7B for subtask 1.** Each node in this DAG (pruned for brevity) is labeled as `<layer index-head index><source residual content><source residual position><head output content>`. The subtask here is to predict *impus* given the generated response *Stella is* (see Appendix G for the complete tokenized input). In this example, there are seven answer writing heads (in green) that are writing *<imp>* to the last residual stream corresponding to *is*. In this particular example, the answer token *imp* is collected from token no. 597 in the input question context by 8 different (in blue).

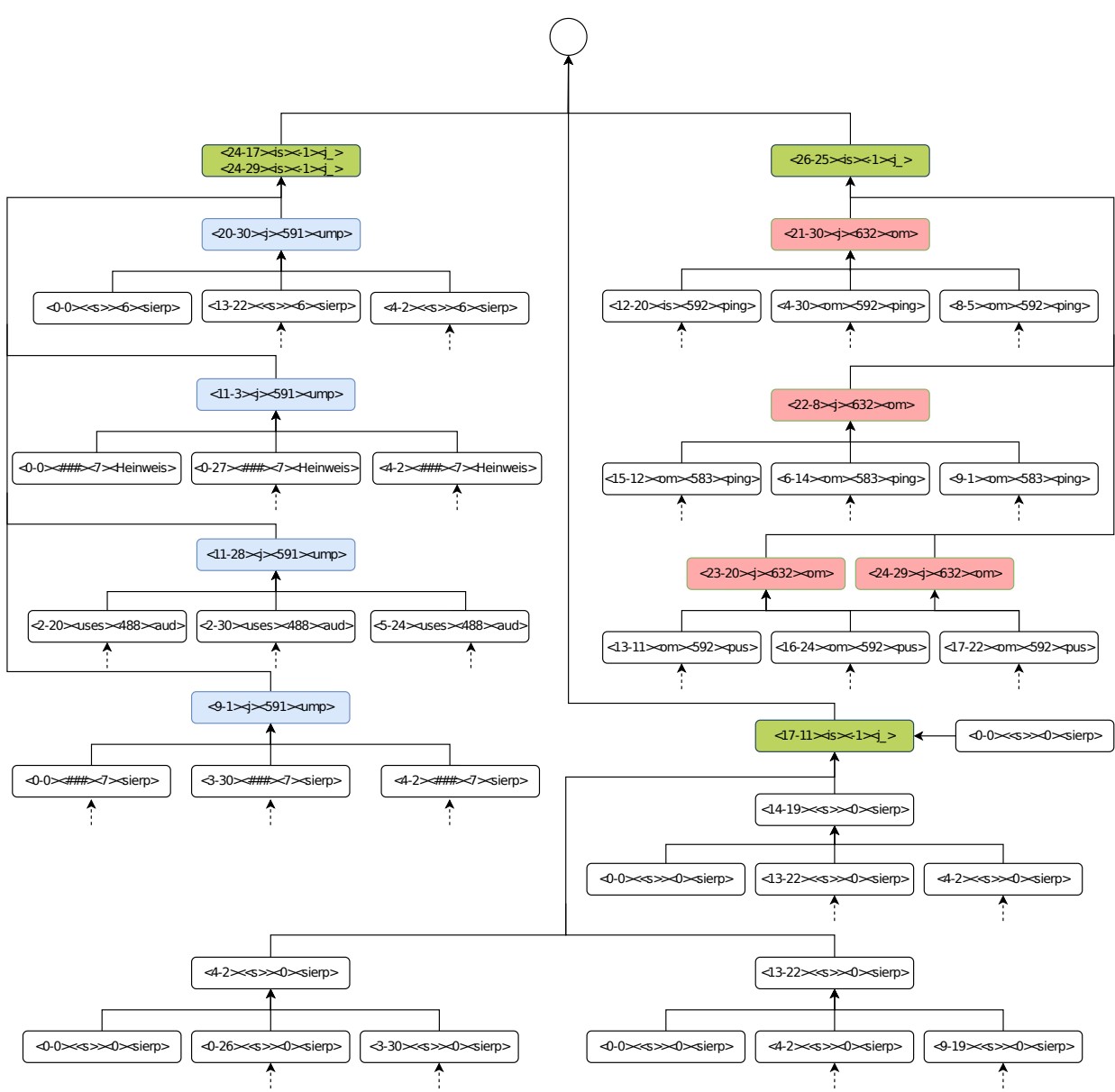

Figure 14: **Infomation flow through attention heads towards answer-writing heads in Llama-2 7B for subtask 5**. The representations of each node are similar to Figure 13. The subtask here is to predict *jompus* given the generated response *Stella is impus. Impus is jompus. Stella is* (see Appendix G for the complete tokenized input). In this example, there are three answer writing heads (in green) that are writing *<j>* to the last residual stream corresponding to *is*. In this particular example, the answer token *imp* is collected from two position, token no. 591 in the input question context 4 heads (in blue), and token no. 632 in the generated context by 4 heads (in red).

