# OpenReview forum: "How to think step-by-step: A mechanistic understanding of chain-of-thought reasoning"
_TMLR — Accepted by TMLR_

### Review · Reviewer_9pmv · 2024-04-21

**Summary Of Contributions:**

This paper analyses the mechanisms involved in the generation of a chain-of-thought for deductive reasoning tasks in a fictional ontology by Llama-2 7B in a few-shot setup. The authors present several interesting results that shed led on the inner-workings of an LLM while performing reasoning tasks with CoT prompting; namely that there is a functional difference between the bottom half of the layers and top half of layers, with the attention heads in the former moving information around and the latter writing answer tokens. Additionally, the authors find that the model utilises multiple pathways for answer writing. The main contribution is shedding light on the inner workings of a model at scale (7B), where most mechanistic interpretability findings in existing literature are about smaller models.

The tasks the authors investigate is a deductive reasoning task that can be naturally split in 3 subtasks: **deciding** which premise to start with, **copying** the relevant tokens from that premise, and **induction** of a new statement. For example, the context might be *"Tumpus is bright. Lempus is tumpus. Max is lempus. True of false: Max is Bright"* and the CoT to generate is then *"Max is lempus. Lempus is tumpus. Max is tumpus. Tumpus is bright. Max is bright. True"*. The contribution here is the insight to use a fictional ontology to be able to solely focus on analysing attention heads, as prior work has shown that the MLPs are mostly used for a key-value-style lookup of facts learned from pretraining (which isn't possible with the fictional ontology).

**More detailed description of method and findings**
The authors use different techniques from mechanistic interpretability to investigate the contribution of specific attention heads to particular subtasks in the deductive reasoning task. They score each attention head in the Transformer three times, and each score reflects the relative importance of that attention head for one of the three subtasks. An attention head can be deemed a "decision head" if the relevant importance score is within a certain threshold. One can "knock out" heads that are not "decision heads", for example, and leave a set of heads that are task-specific. You can do predictions with those on the subtasks required for generating the CoT. Some attention heads are "task-only heads", when they are only one type of task-head (e.g. they are only not knocked out for the induction task). The authors do many analyses in this way to find the following things:
- The three types of heads do not necessarily only get high accuracy on their respective subtasks (e.g. decision heads are not better at decision making than other tasks)
- Induction heads get high accuracy on all the subtasks required for generating the CoT
- The authors hypothesise that induction heads need to mix information, decide which induction to make, and then copy the required tokens, and that's why they are important for all subtasks. They show that indeed the information mixing happens, mostly between layer 10-15
- The authors find that mostly the deeper layers' attention heads are context-abiding (meaning that applying an unembedding layer to their activations leads to a bigram that is actually present in the context)
- The authors further find that there are specific answer-writing heads that directly write the answer to a subtask into the residual stream (also found by applying unembedding to their activations which produces the right answer). There are multiple of these, and they implement different algorithms (found by following their attention patterns to construct a tree)

**Audience:**

Yes

**Broader Impact Concerns:**

N.A.

**Claims And Evidence:**

Yes

**Requested Changes:**

**Requested changes that are necessary for acceptance**
- Figure 2 is quite difficult to parse not only due to my own confusions, but also due to the fact that the 0-9 subtasks are not annotated according to which category the task refers to. Please annotate this somehow in Figure 2, and everywhere else.
- Some of the appendix information needs to be moved to the main text, for example I would propose to write an intuitive summary of the importance scores for each subtask in the main text, because it's kind of hard to interpret the rest of the paper without it. Appendix C.4 also really needs to be moved to the main text.
- Some information simply seems to be missing from the whole paper, like how exactly do you decide which attention heads to keep for Figure 2?
- Figure 3 is also difficult to interpret, especially using a coloring determining accuracy over a histogram of bars. I would suggest either using a different visualisation or at least add a guiding interpretation in the caption text.
- In my experience, writing a strong limitations section only makes a paper's conclusions come across stronger, and I advise the authors to do this as well, especially as there are quite a few differences between their setup and general CoT reasoning by modern LLMs. Additionally, the finding that MLPs are mostly key-value lookups from pretraining (roughly) is not a certainty and I would like to see this discussed in a limitations section.

**Writing style suggestions and other less important suggestions**
- It took me quite a while to parse what the 0-9 subtasks refer to, could you note when you mention them first that they can be seen in Figure 1? Additionally, I would strongly consider using a different word for these 10 subtasks and the 3 subtasks that are task categories, because using subtasks for both is confusing.
- Less hyperboles in the abstract and introduction
- Unimportant nit: I think LLaMA-1 was spelled LLaMA, and Llama-2 is Llama.
- Personal preference: I think the submission could benefit from a representative figure moved up to the top of page 2 with some standalone explanation of the findings

**Strengths And Weaknesses:**

**Strengths**

- The authors apply mechanistic interpretability to a relatively large-scale model (relatively compared to other mech. interp. work), which is an exciting contribution.
- Clever way of focusing on attention heads over MLPs by looking at a dataset with a fictional ontology
- The authors do a lot of analyses and present a lot of relatively well-motivated findings (particularly the results in section 5 and 6 are quite convincing due to the clear patterns found).

**Weaknesses**

- I find the descriptions of the methods in the paper very confusing, and not enough information is provided to interpret the results. I am left with many questions that I don't know the answer to but are necessary to properly follow what is going on and decide whether the claims are sound. Mainly for that reason I decided to write a relatively detailed contributions section, so in case I am misunderstanding something it would be great if the authors can point that out.
- Similarly to the above weakness, there is a lot of information put in the appendix without which you simply cannot understand what is going on in the paper, like the importance scores for the different types of attention heads.
- The paper is studying a relative simple form of reasoning, namely deductive over a fictional ontology, only studies one model family, and only one model size. There is not limitations section present in the paper or a section that discusses how the results might generalise to other model families and model sizes, or what the results might tell us about chain-of-thought reasoning more generally.
- This is a personal preference so feel free to ignore this, but the amount of hyperboles in the first two paragraphs of the introduction is a bit much. Personally, I find that overusing hyperboles makes a text come across less strong.

I'm going to pose my questions below, and I'd like to work with the authors to understand the method better so I can be more convinced of the findings:

Questions:
1. Why do you decide which attention heads to knockout only using the false ontologies?
2. I very much struggle to understand what is going on in section 4. For example, I do not understand how you construct Figure 2. Part of my confusion is caused by the fact that the Y-axis in Figure 2 says accuracy is red and fraction of heads involved is blue, whereas the caption says the opposite. Additionally, in the text you claim subtask 3 is a decision making subtask, but it seems like in Figure 1 subtask 3 is a copy task. Can you tell me if the following is correct for my own understanding and otherwise correct me where I'm wrong:
For example for the decision heads figure (leftmost), you take all the decision subtasks (i.e. 0, 2, 4, 6, 8 in Figure 1), you calculate the importance scores for each attention head, and you select a range of importance scores within which if you keep those attention heads active, you get a relative accuracy of at least 0.9 for the subtask you look at (from 0 to 10). This leads to a different number of attention heads active for each subtask 0-10, and a different absolute accuracy corresponding to the relative accuracy of the decision heads on each those subtasks. This means that if you get an absolutely high accuracy (high blue bar) in the figure for a low fraction of heads involved (low red bar), those attention heads are important for that subtask
3. In Section 5, how are you getting the specific representations that are used as inputs to the classifier? So if you have the following example "A is B, B is C, then A is C", which activations do you take, those when the model is predicting C (the last token)? And which occurrence of B do you take as input to the classifier then, the first or second?
4. why do you call deductive reasoning induction? As far as I know, the type of reasoning the dataset you use requires is purely deductive.

I have more questions but I'm going to start with this and see how far we get in discussion.

---

> ### Author Response · Authors · 2024-04-30
> **Response to reviewer 9pmv (part I)**
>
> Thank you for your thorough review of our paper. We appreciate the insightful questions and constructive feedback provided. Below, we address each of the raised concerns and queries:
>
> **Selection of Attention Heads Using False Ontologies**
>
> We do not decide which attention head to knock out using False ontology. Instead, the mean activation of the model with inputs from False ontology is used as a ‘background noise’ for activation patching. We replace the correct activation of a head (on fictional ontology) with this background noise to stop the information flow from that head. While any form of noise can be used for this purpose (all zero vectors, random vectors sampled from a Gaussian, etc.), we follow the suggestions from Zhang and Nanda [1].
>
> **Clarity of Section 4 and Figure 2**
>
> We acknowledge the mistake made in Figure 2. Blue bars show the accuracy and red bars denote the fraction of heads involved. Subtask 3 is a copy step as you correctly pointed out. Apologies for the errors and the confusion that ensued. Your interpretation of Figure 2 is correct indeed.
>
> *Changes in manuscript (in blue)*:
>
> i) Figure 2
>
> ii) Section 5 page 8 -> “For example, in subtask 2, since we assumed it to be a decision-making subtask, we took only those decision-making heads that were not present in the copying or inductive reasoning heads”
>
> **Clarity of Figure 3**
>
> We have updated the caption to Figure 3 with a detailed explanation.
>
> *Changes in manuscript (in blue)*:
>
> i) Figure 3 caption
>
> **Representation Input for Classifier in Section 5**
>
> We extract the hidden states corresponding to different tokens associated with the fictional entities at each layer while predicting the last token (note that due to the autoregressive attention, the representations associated with past tokens do not change upon the arrival of later tokens). To curate the dataset we form pairs of positive, negative, and unrelated entities. For example, given the following input: A is B. B is not C. D is F, we construct the following:
>
> A <-> B  (positive pairs)
>
> B <-> C (negative pairs)
>
> A <-> D, B <-> D, C <-> D, A <-> F, B <-> F, C <-> F (unrelated pairs)
>
> There can be multiple occurrences of a token in the input context (e.g., there are two occurrences of [B] in the above example). As a result, there will be multiple residual streams emanating from the same token. To decide which residual streams to pair with (in positive/negative pairing), we rely on immediate occurrence. For example, in the above input, the first occurrence of B is positively paired with A, while the second B is negatively paired with C. We do not construct positive or negative pairs without immediate cooccurrence.
> More information is also provided in Appendix D.
>
> *Changes in manuscript (in blue)*:
>
> Section 5 page 8 -> “There can be multiple occurrences of a token in the input context (e.g., there are two occurrences of [B] in the above example). As a result, there will be multiple residual streams emanating from the same token. To decide which residual streams to pair with (in positive/negative pairing), we rely on immediate occurrence. For example, given the input [A] is [B]. [A] is not [C], we take the first occurrence of [A] to be positively related to [B], while the second occurrence of [A] as negatively related to [C].”
>
>
>
> [1] Zhang, F., & Nanda, N. (2023). Towards best practices of activation patching in language models: Metrics and methods. arXiv preprint arXiv:2309.16042.

---

> ### Author Response · Authors · 2024-04-30
> **Response to reviewer 9pmv (part II)**
>
> **Terminology Clarification Regarding "Induction"**
>
> Deductive reasoning typically involves deriving specific conclusions from generalized premises, whereas inductive reasoning refers to reaching generalized truths from a set of observations. In our setting, the reasoning step required to predict “Max is lempus” from the prior context “Max is lempus. Lempus is Dumpus.” follows from similar examples observed in the few-shot context. There is no generalized premise that elicits such a step because there is no notion of transitivity stated explicitly in the ontological relationships.
>
> **Attention heads chosen from Figure 2**
>
> Inductive reasoning heads perform consistently in each of the tasks, with the highest accuracy and a lesser fraction of heads involved. For this reason, we kept the inductive reasoning moving further in the paper.
>
> *Changes in manuscript (in blue)*:
>
> i) Section 4 page 8 -> “Figure 3 shows the accuracy of the model at each different subtask indices with heads pruned at different \(\delta(h_{j,k})\) (we show the histogram over \(\mu(h_{j,k})\) and start pruning heads from the central bin corresponding to the smallest values of \(\delta(h_{j,k})\)). Among the $1,024$ heads of Llama-2 7B, we can see that pruning the model to as low as $\sim$400 heads retains \(>\)90\% of the original accuracy (see Table 1 in Appendix for subtask-wise statistics).”
>
> ii) Figure 2 caption
>
>
> **Absence of limitation section**
>
> We acknowledge the absence of a limitation section in the paper and we have added one in the updated version of the paper.
>
> *Changes in manuscript (in blue)*:
>
> Section 7 page 14-15 -> “The design of this study imposes certain limitations in its scope. First and foremost, we analyze a very specific type of reasoning problem. While the fictional ontology with a restricted CoT template provides ease of analysis, free-form reasoning can introduce further complex dynamics not captured in this study. We also could not address the role of MLPs in reasoning. While existing literature points to their role as factual memory, it should be noted that the model also memorizes a diverse set of token-token associations pertaining to the structure of language and not just factual associations. Such associations are likely to play a decisive role in a grounded reasoning setup that our analysis ignores. Finally, we use a few-shot prompting regime to ensure that the model follows a specific structure of reasoning steps. With zero-shot CoT, more complex mechanisms are bound to emerge.”
>
>
>
> **Confusion around the sub-task annotation**
>
> We have incorporated the subtask-wise annotation in Figure 1 in the updated manuscript. Please let us know if this resolves the confusion.
>
> *Changes in manuscript*:
>
> i) Figure 1
>
> **Fewer hyperboles in the abstract and introduction**
>
> We have updated the abstract and the introduction with less hyperbole following your suggestions.
>
> *Changes in manuscript (in blue)*
>
> i) Abstract -> “We observe a functional rift in the middle layers of the LLM. Token representations in the initial half remain strongly biased towards the pretraining prior, with the in-context prior taking over in the later half. This internal phase shift manifests in different functional components: attention heads that write the answer token appear in the later half, attention heads that move information along ontological relationships appear in the initial half, and so on.”
>
> ii) Introduction, page 1 -> “Recent advancements with Large Language Models (LLMs) demonstrate their remarkable reasoning prowess with natural language across a diverse set of problems (OpenAI, 2024; Kojima et al., 2022; Chen et al., 2023). Yet, existing literature fails to describe the neural mechanism within the model that implements those abilities – how they emerge from the training dynamics and why they are often brittle against even unrelated changes. One of these capabilities of LLMs that has boosted their potential in complex reasoning is Chain-of-Thought (CoT) prompting (Wei et al., 2022b; Kojima et al., 2022). Instead of providing a direct answer to the question, in CoT prompting, we expect the model to generate a verbose response, adopting a step-by-step reasoning process to reach the answer. Despite the success of eliciting intermediate computation and their diverse, structured demonstration, the exact mechanism of CoT prompting remains mysterious. Prior attempts have been made to restrict the problem within structured, synthetic reasoning to observe the LLM’s CoT generation behavior (Saparov & He, 2023); context perturbation towards causal modeling has also been used (Tan, 2023). However, these endeavors produce only an indirect observation of the mechanism; the underlying neural algorithm implemented by the LLM remains in the dark.”
>
> **LLaMA-2 vs Llama-2**
>
> We have replaced the nomenclature accordingly across the manuscript.
>
> We welcome your further suggestions to improve the quality of the manuscript.

---

> > ### Comment · Reviewer_9pmv · 2024-05-16
> > **Thanks for the detailed responses**
> >
> > I'd like to thank the authors for the detailed responses and taking the time to answer all my questions extensively + updating the manuscript to clarify confusions. Most of my required changes have been addressed properly. Some issues remain though, which I'll detail below.
> >
> > - Thanks for updating Figure 2! I'd still however suggest to annotate the subtasks in figure 2, so you don't need to jump between 1 and 2 and keep them in mind. You could just for example in each figure make the relevant bars darker colored or something (or striped), so for example for leftmost figure 2, somehow make the bars for 0, 2, 4, 6, 8 stand out, or bolden the labels on the x-axis.
> > - Small error: twice "model" in caption fig 3 ("of the pruned **model model** relative").
> > - Thanks for adding the limitations section! Could you speculate on to what extent your results say something about CoT reasoning in the wild based on these limitations?

---

> > > ### Author Response · Authors · 2024-05-18
> > > **Additional changes**
> > >
> > > Thanks a lot for your suggestions. We have tried to incorporate them in the revised version (changes highlighted in blue).
> > >
> > > **Highlight relevant tasks in Figure 2**
> > >
> > > We have made the subtask indices in the x-axis for each subplot relevant to the subtask category in boldfaced green fonts. Please let us know if this follows your suggestion successfully.
> > >
> > > *Changes in manuscript*
> > >
> > > Figure 2 and Figure 2 caption
> > >
> > > **Typo in Figure 3 caption**
> > >
> > > We have corrected the typo. Thanks again for pointing it out!
> > >
> > > **Implication of our findings based on the limitations**
> > >
> > > We have updated the paragraph *Implications for future research* by putting it after the *Limitations* paragraph and incorporating the suggested implications.
> > >
> > > *Changes in manuscript (in blue)*
> > >
> > > Section 7 page 15 -> "Although real-world reasoning problems are far more dynamic than our strictly controlled setup of ontology-based reasoning, many of the results hold in the case of CoT in the wild. The existence of multiple neural pathways of answer generation is intrinsic to the model (although in real-world reasoning, such pathways will contain MLP blocks as a source of the answer, instead of the context only). The token-mixing mechanism that we uncovered remains true for any ontological relationship, not just fictional. Additionally, the more out-of-distribution (with respect to pretraining data) the reasoning problem becomes, the stronger the rift in terms of context abidance we will observe."

---

> ### Comment · Reviewer_9pmv · 2024-05-16
> **Deductive vs inductive reasoning**
>
> The fact that the deductive reasoning task has to be inferred from few-shot examples does not make the it a inductive reasoning in my opinion, I think the task remains textbook deductive reasoning, but I can let this go.

---

### Review · Reviewer_Fxq8 · 2024-04-26

**Summary Of Contributions:**

The authors report the results of a series of experiments to better understand characteristics of computations taking place during chain-of-thought reasoning with an LLM (LLaMA-2). The experiments are performed with questions requiring to reason about "fictional" characters and facts to avoid contaminating the answers with pre-learned broad knowledge of facts and only rely on the general reasoning abilities of the LLM. A notable result is they do not find a decomposition of the required "natural" decomposition of the reasoning problem into tasks that mapped to any subset of heads: instead heads seemed to participate in many tasks. However, they found that nonlinear probes (but not linear probes) could detect the required movement of information and observed important differences between early and late layers.

**Audience:**

Yes

**Claims And Evidence:**

Yes

**Requested Changes:**

Please have the text revised by a human or an AI system to correct numerous little grammatical errors.

See the weak points made in the above section, which should be addressed.

accuracy of 0.9: clarify if it is 90%.

**Strengths And Weaknesses:**

I find particularly interesting the finding regarding the difficulty of identifying subsets of heads associated with a particular subtask (but that is according to a particular decomposition of the reasoning task, it does not exclude that other ways of thinking about the problem would not find such an assignment).

I find the other results more difficult to interpret. Nonetheless, this is part of a growing body of observations to make sense of what is going on in Transformers trained as LLMs, with a focus on reasoning, which is original and interesting.

The paragraph on the fictional ontology took me a while to understand. The main message should be that we set up a set of invented facts and entities on which the LLM has to reason. However, these should not contradict the facts seen by the LLM during training. They should be hypothetical facts which could be true in principle and thus indeed constitute a fiction. However, if they contradicted the training set facts, it would be a problem in that those training set facts would bias the reasoning done by the LLM. Also in this paragraph we introduce the term "substasks Ti" without really explaining what is intended. An example at this point would be neeeded.

---

> ### Author Response · Authors · 2024-04-30
> **Response to reviewer Fxq8**
>
> Thank you for your thorough review of our paper. We appreciate the insightful questions and constructive feedback provided. Below, we address each of the raised concerns and queries:
>
> **Implications of fictional ontology**
>
> We have made the changes you suggested in the manuscript. Note that fictional ontology does not interfere with the pretrained memorization. It is the false ontologies that require the model to actively act against the memorized factual associations.
>
> *Changes in manuscript (in blue)*:
>
> Section 3 page 4 -> “A fictional ontology ensures zero interference between the entity relationships presented in the question and the world-knowledge acquired by the model in the pertaining stage. This further minimizes the involvement of the MLP layers in the neural mechanisms implemented by the model as they typically serves as the parametric fact memory of an LM”
>
> **Unused notations**
>
> We have removed the mentioned notation to avoid confusion.
>
> *Changes in manuscript (in blue)*:
>
> Section 3 page 4 -> “With CoT, one then needs to repeat such perturbation process for all the constituent subtasks.”
>
> **Clarification on accuracy**
>
> Yes, an accuracy of 0.9 here signifies 90% accuracy.
>
> We welcome your further suggestions to improve the quality of the manuscript.

---

### Review · Reviewer_KCnw · 2024-05-03

**Summary Of Contributions:**

This work seeks for a mechanistic understanding of chain-of-thought (CoT) reasoning in large language models (LLMs). To this end, this work leverages Llama-2 7B in few-shot prompting for multi-step reasoning over fictional ontologies. Various findings have been reported after the analysis. Empirical studies show that  LLMs deploy multiple parallel pathways of answer generation for step-by-step reasoning. These parallel pathways provide sequential answers from the input question context as well as the generated CoT. Besides, there is a functional rift at the very middle of the LLM (16th decoder block in case of Llama-2 7B), which marks a phase shift in the content of residual streams and the functionality of the attention heads.

**Audience:**

Yes

**Claims And Evidence:**

No

**Requested Changes:**

1. Check the claims and accurately position the contribution of this work.

2. Justify the choice of using the vanilla Llama-2 7B for analysis. Possibly consider a more convincing analysis on CoT-tuned LLMs, such as Llama-2 Alpaca or Vicuna.

**Strengths And Weaknesses:**

Strengths:

1. The research topic of understanding CoT is very important and has attracted increasing interest in the community. This work presents a comprehensive analysis of the mechanism of CoT reasoning in LLMs.

2. The empirical results are comprehensive. Some results well align with the observations in related CoT studies.

Weaknesses:

1. There are possible over-claims.  Various prior studies have tried to understand CoT reasoning either theoretically [1] or empirically [2][3]. Though the analysis approach may vary, it is risky to claim that "To the best of our knowledge, this is the first attempt towards mechanistic investigation of CoT reasoning in LLMs.”

2. There are some findings that may not well generalize to other LLMs. This work ambitiously aims for a mechanistic understanding of
chain-of-thought reasoning in LLMs. However, the analysis is purely based on the Llama-2 7B model (w/ few-shot prompting), which is not widely adopted or optimized for real CoT-based tasks. Besides, there are some "magic" observations, e.g., "a functional rift at the very middle of the LLM (16th decoder block in case of Llama-2 7B)" It is not clear whether those findings could be different in other LLMs. I understand the potential computation cost when using a larger model. However, as the vanilla  Llama-2 7B is not a strong baseline for CoT reasoning, it would be more convincing if CoT-tuned Llama (e.g., Alpaca or Vicuna series)  could be adopted to support the findings.

3. The task definition is not clear. This paper mentions subtasks 0-9 many times. Are all the tasks the same as those in Figure 1, say, Decision-type subtasks (0, 2, 4, 6, and 8), Copy-type subtasks (1, 3, and 7), and Induction-type subtasks (5 and 9)? A clear categorization of those tasks would be useful.

References:

[1] Ben Prystawski and Noah D Goodman. Why think step by step? Reasoning emerges from the locality of experience. Advances in Neural Information Processing Systems, 2023.

[2] Xinyi Wang and William Yang Wang. Reasoning ability emerges in large language models as aggregation of reasoning
paths: A case study with knowledge graphs. In Workshop on Efficient Systems for Foundation Models@ ICML2023,
2023.

[3]Zhen Bi, Ningyu Zhang, Yinuo Jiang, Shumin Deng, Guozhou Zheng, Huajun Chen. When Do Program-of-Thought Works for Reasoning?. In Proceedings of the AAAI Conference on Artificial Intelligence (Vol. 38, No. 16, pp. 17691-17699), 2024.

---

> ### Author Response · Authors · 2024-05-06
> **Response to reviewer KCnw**
>
> Thank you for your thorough review of our paper. We appreciate the insightful questions and constructive feedback provided. Below, we address each of the raised concerns and queries:
>
> **Possible over-claim**
>
> Thank you for pointing out the prior papers investigating chain-of-thought reasoning. We have incorporated them in the discussion of related work in the updated manuscript. We would like to point out that all three of the mentioned papers investigate the reasoning capabilities of LLMs from perspectives that are fundamentally different from ours. Prystawski et al. [1] explain the emergence of CoT using localized statistical structures of the training data. The findings of Wang and Wang [2] empirically validate the need for such local structures realized over knowledge graphs. Bi et al. [3] investigate the effectiveness of code data in eliciting reasoning in LLMs. Our work does not investigate the emergence of reasoning (or CoT), but rather the neural algorithm implemented by the model while performing CoT generation. For this very reason, we think that our claim "To the best of our knowledge, this is the first attempt towards mechanistic investigation of CoT reasoning in LLMs.” stands.
>
> *Changes in manuscript (in blue)*
>
> Section 2 page 3 -> Prystawski et al. (2024) theoretically associated the emergence of CoT with localized statistical structures within the pertaining data. Wang & Wang (2023) investigated reasoning over knowledge graphs to characterize the emergence of CoT. They identified that reasoning capabilities are primarily acquired in the pertaining stage and not downstream fine-tuning while empirically validating Prystawski et al. (2024)’s theoretical framework in a more grounded manner. Bi et al. (2024) found that while reasoning via program generation, LLMs tend to favor an optimal complexity of the programs for best performance.
>
>
>
> **Choice of model**
>
> We experimented with Vicuna (instruction finetuned on Llama-2) and Llama-2 alpaca. Unfortunately, they proved to be much worse in reasoning on fictional and false ontologies compared to the base Llama-2. Following is a comparison between the three models:
>
> | Model              | Llama-2-7B (base) | Llama-2-7B (alpaca) | Vicuna-7B |
> |--------------------|-------------------|---------------------|-----------|
> | PrOntoQA fictional | 35.9              | 24.4                | 14.95     |
> | PrOntoQA false     | 52.5              | 13.82               | 13.75     |
>
> Among the prior works you suggested, both Prystawski et al. [1] and Wang and Wang [2] suggest that reasoning capabilities in LLMs are acquired in the pretraining stage. Also, multiple recent research claim that the neural mechanisms of an LLM remain mostly intact with generalized finetuning or alignment tuning [4, 5]. Prakash et al. [5] associate the superior performance boosts achieved via finetuning with the ability to handle augmented positional information. In this light, we think our analysis of the base variant of Llama-2 will hold across different models.
>
> **Confusion over subtask template**
>
> As we mentioned in Section 4 (first paragraph on page 6), all the analysis in our paper follows the structure presented in Figure 1. Furthermore, we have provided clear indexing of the tasks in the updated version of Figure 1 in the revised manuscript.
>
> *Changes in manuscript*
>
> Figure 1
>
> We welcome your further suggestions to improve the quality of the manuscript.
>
> [1] Ben Prystawski, Michael Y. Li, and Noah D Goodman. Why think step by step? Reasoning emerges from the locality of experience. Advances in Neural Information Processing Systems, 2023.
> [2] Xinyi Wang and William Yang Wang. Reasoning ability emerges in large language models as aggregation of reasoning paths: A case study with knowledge graphs. In Workshop on Efficient Systems for Foundation Models@ ICML2023, 2023.
> [3] Zhen Bi, Ningyu Zhang, Yinuo Jiang, Shumin Deng, Guozhou Zheng, Huajun Chen. When Do Program-of-Thought Works for Reasoning?. In Proceedings of the AAAI Conference on Artificial Intelligence (Vol. 38, No. 16, pp. 17691-17699), 2024.
> [4] Samyak Jain, et al. Mechanistically analyzing the effects of fine-tuning on procedurally defined tasks. ICLR 2024.
> [5] Nikhil Prakash, et al. Fine-Tuning Enhances Existing Mechanisms: A Case Study on Entity Tracking. ICLR 2024.

---

### Decision · Action_Editor_oL6N · 2024-06-25

**Recommendation:** Accept as is

**Comment:**

Overall, the reviewers were satisfied with the paper's methodology and results. Mechanistic interpretability (MI) has so far been applied to relatively small-scale models; this is the first application of MI to an LLM.  The reviewers had some concerns about the level of methodological detail and the discussion of related work in the original paper. However, the authors have done a good job of revising the paper and have adequately addressed these concerns. Given this, I recommend acceptance.

**Audience:**

Mechanistic interpretability is an exciting new direction of ML research, and this is the first time the approach has been applied to an LLM. Considering this, the result should interest a nontrivial subset of the TMLR audience.

**Claims And Evidence:**

The paper applies a mechanistic interpretability lens to LLMs that do chain-of-thought generation. Through an analysis of a Llama2 7B model, the paper concludes that: (i) LLMs use multiple parallel pathways to chain-of-thought answer generation, and (ii) there is a phase transition, from the pretraining prior to an in-context prior, around the middle layers of the LLM. In the reviewers' and my opinions, the evidence the paper gives to back up these claims is sound.